# Plug-and-Play: An Efficient Post-training Pruning Method for Large Language Models

**Yingtao Zhang**[1,2,*] **Haoli Bai**[4], **Haokun Lin**[5], **Jialin Zhao**[1,2], **Lu Hou**[4],
**Carlo Vittorio Cannistraci**[1,2,3]
[1]Center for Complex Network Intelligence, Tsinghua Laboratory of Brain and Intelligence
[2]Department of Computer Science, Tsinghua University
[3]Department of Biomedical Engineering, Tsinghua University
[4]Huawei Noah's Ark Lab, [5]Institute of Automation, Chinese Academy of Sciences
Corresponding to {zhangyingtao1024,kalokagathos.agon}@gmail.com

## Abstract

With the rapid growth of large language models (LLMs), there is increasing demand for memory and computation in LLMs. Recent efforts on post-training pruning of LLMs aim to reduce the model size and computation requirements, yet the performance is still sub-optimal. In this paper, we present a plug-and-play solution for post-training pruning of LLMs. The proposed solution has two innovative components: 1) **Relative Importance and Activations** (RIA), a new pruning metric that jointly considers the weight and activations efficiently on LLMs; and 2) **Channel Permutation**, a new approach to maximally preserve important weights under N:M sparsity. The two proposed components can be readily combined to further enhance the N:M semi-structured pruning of LLMs. Our empirical experiments show that RIA alone can already surpass all existing post-training pruning methods on prevalent LLMs, e.g., LLaMA ranging from 7B to 65B. Furthermore, N:M semi-structured pruning with channel permutation can even outperform the original LLaMA2-70B on zero-shot tasks, together with practical speedup on specific hardware. Our code is available at: https://github.com/biomedical-cybernetics/Relative-importance-and-activation-pruning

## 1 Introduction

Recent research on large language models (LLMs) has attracted significant interest. These LLMs, characterized by their vast number of parameters, have exhibited remarkable proficiency across a wide range of tasks. However, deploying such models poses challenges due to their substantial size, computational demands, and execution time. To address this, several methods for network compression have been explored, such as model quantization (Bai et al., 2020; Frantar et al., 2022; Xiao et al., 2023; Lin et al., 2023) and network pruning (LeCun et al., 1989; Hassibi et al., 1993; Mocanu et al., 2018; Sun et al., 2023; Frantar & Alistarh, 2023).

Unlike quantization techniques, which adjust the precision of weights or activations to compress the network, network sparsity primarily targets the elimination of redundant or useless weights within models. Despite its potential, the exploration of sparsity in LLMs remains limited. Generally, neural networks can achieve sparsity through three primary methods: 1) sparse training (Lee et al., 2018; Mocanu et al., 2018; Evci et al., 2020; Sanh et al., 2020; Yuan et al., 2021; Hoang et al., 2022; Zhang et al., 2023); 2) pruning-aware training (Han et al., 2015; Liu et al., 2021); and 3) post-training pruning (PTP) (Hassibi et al., 1993; Li & Louri, 2021; Frantar & Alistarh, 2023; Sun et al., 2023). However, both sparse training and during-training pruning require multiple rounds of iterative training, which is computationally costly and time-consuming, especially for LLMs. Therefore, PTP on a well-pre-trained model represents a more reasonable approach for LLMs.

The primary challenge associated with post-training pruning lies in the substantial performance degradation compared to dense models. Current methods, like SparseGPT (Frantar & Alistarh,

---

[*]This work is partially done during the internship at Huawei Noah's Ark Lab.

2023) and Wanda (Sun et al., 2023), exhibit promising results in unstructured pruning. Nonetheless, to achieve practical speed-up, it is favored to conduct N:M semi-structured pruning, which can be supported by specific hardware with sparse matrix multiplications (Mishra et al., 2021). These prior methods (Sun et al., 2023; Frantar & Alistarh, 2023) under the N:M sparsity suffer from significant performance drops, thereby restricting their applications in practice. Recently, (Pool & Yu, 2021) has introduced an input channel permutation method using the greedy search, which can boost the performance under the N:M sparsity. However, the greedy search is time-consuming on LLMs and thus is not feasible.

In this paper, we introduce a plug-and-play post-training pruning method for LLMs. Specifically, the method comprises two key components. First, we introduce **Relative Importance and Activation** (**RIA**), a new pruning metric for LLM pruning. We show that prior pruning metrics (Frantar & Alistarh, 2023; Sun et al., 2023) tend to prune away entire channels of network weights, which is undesirable for both unstructured and N:M semi-structured pruning. Instead, RIA jointly considers the input and output channels of weight and the activation information, effectively mitigating such issues. Second, we also consider a better way to convert LLM weight matrices to adhere to N:M sparsity patterns. Unlike existing methods that directly convert the weight matrix to N:M sparsity, we propose **channel permutation** to properly permute the weight channels so as to maximally preserve the important weights under N:M structures. Finally, the proposed RIA and channel permutation can be readily combined, leading to an efficient and plug-and-play approach for the real-world acceleration of sparse LLMs inference. We name the proposed method "plug-and-play" since 1) it does not need additional fine-tuning or retraining; 2) it can be adopted by any models that have linear layers; 3) it has negligible performance drop when applying channel permutation in the zero-shot experiments.

Extensive evaluation of open-sourced LLMs (e.g., LLaMA (Touvron et al., 2023a), LLaMA-2 (Touvron et al., 2023b), and OPT (Zhang et al., 2022a)) demonstrates that RIA outperforms SOTA one-shot PTP methods in both unstructured sparsity and N:M sparsity scenario. Additionally, channel permutation can be efficiently scaled to LLMs with over 70B parameters within 2 hours, yet recover a lot of the performance drop caused by N:M constraint and surpass the performance of dense models in 3 out of 5 evaluated zero-shot datasets. By employing RIA and channel permutation, LLMs can undergo a smooth transition into N:M constraints. This ensures hardware compatibility while maintaining the intact performance of the pruned model.

## 2 RELATED WORK

**Post-Training Pruning.** Post-training pruning (PTP) methods trace their roots to OBD (LeCun et al., 1989), which employs the hessian matrix for weight pruning. OBS (Hassibi et al., 1993) further refines the approach by adjusting remaining weights to minimize loss changes during pruning. The advent of Large Language Models (LLMs) has sparked interest in leveraging the Hessian Matrix for pruning, exemplified by works like AdaPrune (Li & Louri, 2021) and Iterative Adaprune (Frantar & Alistarh, 2022) targeting BERT (Devlin et al., 2018). However, weight reconstruction via Hessian Matrix inverse incurs substantial computational complexity at $O(N^4)$. SparseGPT (Frantar & Alistarh, 2023) reduces this complexity to $O(N^3)$, while Wanda (Sun et al., 2023) leverages input activations for efficient one-shot PTP with reduced pruning time and comparable performance to SparseGPT. Our proposed method, Relative Importance and Activations (RIA) inherits the time-saving advantages of Wanda while simultaneously enhancing the performance of the pruned LLMs. Note that the method proposed in this article is designed for model compression without retraining and finetuning, which differentiates it from Dejavu (Liu et al., 2023), which requires additional training steps. Such post-training methods are also preferred for model quantization as well (Bai et al., 2022; Xiao et al., 2023; Lin et al., 2023; Liu et al., 2024).

**N:M Sparsity.** Recently, NVIDIA has introduced the N:M constraint sparsity (Mishra et al., 2021) as a method to compress neural network models while preserving hardware efficiency. This N:M sparsity constraint stipulates that at least N out of every contiguous M element must be set to zero, thereby accelerating matrix-multiply-accumulate instructions. For instance, a 2:4 constraint ratio results in 50% sparsity, effectively doubling the model's inference speed when utilizing the NVIDIA Ampere GPU architecture. However, directly applying SOTA unstructured pruning methods to meet the N:M sparsity often leads to a noticeable decline in performance, as demonstrated in Table 6. Some approaches (Hubara et al., 2021; Zhou et al., 2021; Zhang et al., 2022b) suggest fine-tuning

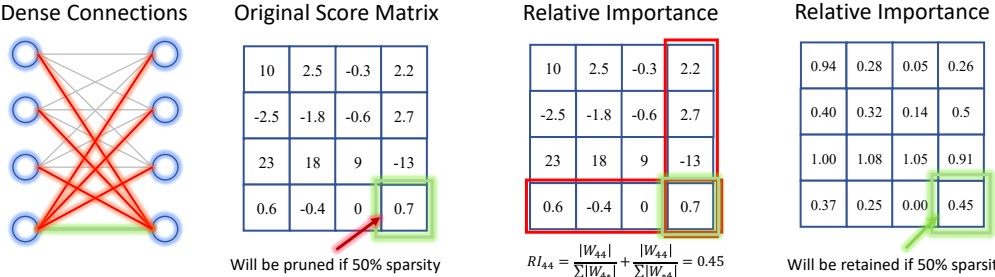

Figure 1: An example of Relative Importance. The connection at position 44 will be removed based on global network magnitude. However, it is retained when evaluated for its significance within relative connections.

pruned models to recover capacity, but this is prohibitively expensive for Large Language Models (LLMs).

**Matrix Permutation.** N:M sparsity primarily targets the application of sparsity in the input channel dimension. (Pool & Yu, 2021) presents a permutation method to identify the optimal permutation of input channels through an exhaustive greedy search and an escape phase to navigate local minima. However, this greedy approach becomes impractically time-consuming when applied to LLMs due to their extensive linear layers. In this study, we leverage the specific characteristics of LLMs and propose a Channel Permutation strategy to reduce the computational overhead efficiently.

## 3  LLM PRUNING WITH RELATIVE IMPORTANCE AND ACTIVATIONS

### 3.1  POST-TRAINING PRUNING: PRELIMINARIES

Post-training pruning (PTP) typically starts from the pre-trained network, removes redundant parameters, and does not need end-to-end fine-tuning. Unlike training-based pruning methods (Mocanu et al., 2018; Sanh et al., 2020; Zhang et al., 2023; Tao et al., 2023; Lin et al., 2024), PTP is fast, resource-saving, and therefore preferred for compressing LLMs (Frantar & Alistarh, 2023; Sun et al., 2023). PTP is currently prevalent in *unstructured pruning* and *N:M semi-structured pruning*, which is also the main focus of this paper. It is less applied in structured pruning (Ma et al., 2023) due to a larger performance drop.

A common approach to achieve PTP is layer-wise pruning, e.g., minimizing the discrepancy square error between the dense and pruned model layer-by-layer recursively. Specifically, we denote the input of the $l$-th linear layer as $\mathbf{X}_l$, and weight $\mathbf{W}_l \in \mathbb{R}^{r \times c}$, where $r$ and $c$ represent the number of output and input channels respectively. Our primary goal is to find the pruning mask $\mathbf{M}_l \in \{0, 1\}^{r \times c}$ that minimizes the $\ell_2$ distance error between the original and pruned layer.

Therefore, the objective can be formally expressed as follows:

$$\arg\min_{\mathbf{M}_l} ||\mathbf{W}_l\mathbf{X}_l - (\mathbf{M}_l \odot \mathbf{W}_l) \cdot \mathbf{X}_l||_2^2, \quad \text{s.t..} \quad ||\mathbf{M}_l||_0 \leq k, \tag{1}$$

where $k$ represents the number of remaining weights determined by the pruning ratio, and $|| \cdot ||_0$ is the $\ell_0$-norm (e.g., the number of non-zero elements). To solve $\mathbf{M}_l$ in Equation 1, there are various pruning metrics, e.g., magnitude-based pruning, that mask the weight below a certain threshold. Nonetheless, we show that these prior pruning metrics have intrinsic drawbacks, as discussed in the following section. Optionally, the weight $\mathbf{W}_l$ in Equation 1 can also be reconstructed via closed-form updates (Frantar & Alistarh, 2023); see Appendix D for more discussions.

### 3.2  RELATIVE IMPORTANCE: A NEW PRUNING METRIC

We present the relative importance (dubbed as **RI**), a new metric for LLM pruning. We find that prevalent PTP methods tend to suffer from *channel corruption*, i,e., the entire input or output channel is pruned away. This is akin to node removal in network science terms, and it significantly diminishes the performance of Large Language Models (LLMs), in a manner similar to structured pruning. To

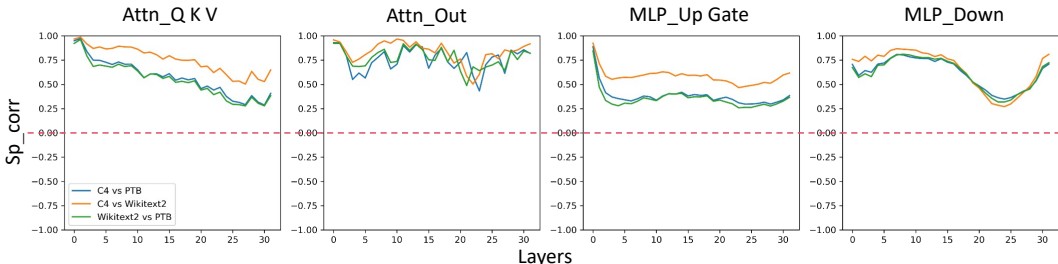

Figure 2: Spearman's Rank correlation of LLaMA2-13B activations from Wikitext2, C4, and PTB, each with 128 samples. Note that the Q, K, and V layers all share the same input activations. Similarly, the Up and Gate layers also receive identical input activations.

illustrate this, Figure 1 shows a linear layer with $\mathbf{W} \in \mathbb{R}^{4 \times 4}$. Magnitude-based pruning with 50% unstructured sparsity will corrupt $\mathbf{W}_{4*}$, i,e, the 4-th output channel. In practice, we find similar issues also exist in other prevalent pruning metrics, e.g., Wanda (Sun et al., 2023) prunes around 500 channels out of 5120 channels in some layers, with more than 10% channels corrupted. Given that well-trained LLMs contain unique information in the input and output channels, it is critical to avoid channel corruption in post-training pruning. We explain the phenomenon of channel corruption in detail in Appendix E.

To mitigate such issues, the proposed relative importance (RI) aims to re-evaluate the importance of each weight element $\mathbf{W}_{ij}$ based on all connections that originate from the input neuron $i$ or lead to the output neuron $j$. Specifically, the relative importance for $\mathbf{W}_{ij}$ can be calculated as:

$$\mathbf{RI}_{ij} = \frac{|\mathbf{W}_{ij}|}{\sum |\mathbf{W}_{*j}|} + \frac{|\mathbf{W}_{ij}|}{\sum |\mathbf{W}_{i*}|}, \tag{2}$$

where $\sum |\mathbf{W}_{*j}|$ sums over the absolute values of the weights in input channel $j$, and similarly $\sum |\mathbf{W}_{i*}|$ for the sum of the weights in output channels $i$. The resulting score $\mathbf{RI}_{ij}$ offers insight into the relative importance of weight $\mathbf{W}_{ij}$ in the context of its connections to neurons $i$ and $j$.

### 3.3 INCORPORATING ACTIVATIONS INTO RELATIVE IMPORTANCE

The proposed relative importance can be further combined with activations to assess better the weight significance, dubbed as relative importance and activation (**RIA**). Recent findings (Xiao et al., 2023) show that the occurrence of activation outliers has become a well-known issue in quantizing LLMs. Our visualizations on LLaMA and OPT also confirm these outliers; see Figure 7 and Figure 8 for details.

Moreover, we find that activation outliers persist regardless of the dataset or parts of the model. To see this, we calculate the Spearman's Rank Correlation Coefficient of activations between different datasets. From Figure 2, it can be observed that pairwise correlations of activations exhibit positive values and similar trends across different layers and Transformer modules. We offer evidence that the Spearman Rank correlation between pairs of the activations of different datasets is positive. This positivity is a necessary condition to incorporate the activation into our RIA formula. And indeed it is always satisfying.

Built upon Equation 2, for each element $\mathbf{W}_{ij}$, RIA further combines $\ell_2$-norm of activations $||\mathbf{X}_i||_2$ as follows:

$$\mathbf{RIA}_{ij} = \mathbf{RI}_{ij} \times (||\mathbf{X}_i||_2)^a = \left( \frac{|\mathbf{W}_{ij}|}{\sum |\mathbf{W}_{*j}|} + \frac{|\mathbf{W}_{ij}|}{\sum |\mathbf{W}_{i*}|} \right) \times (||\mathbf{X}_i||_2)^a, \tag{3}$$

where a power factor $a$ is introduced to control the strength of activations. Our empirical results in Figure 9 show that $a = 0.5$ works generally well for different LLMs.

## 4 TURNING INTO N:M SPARSITY

This section studies how to turn LLMs into N:M sparsity with the pruning metric. N:M sparsity is usually favored by post-training pruning given its practical speed-up on specific hardware. Unlike

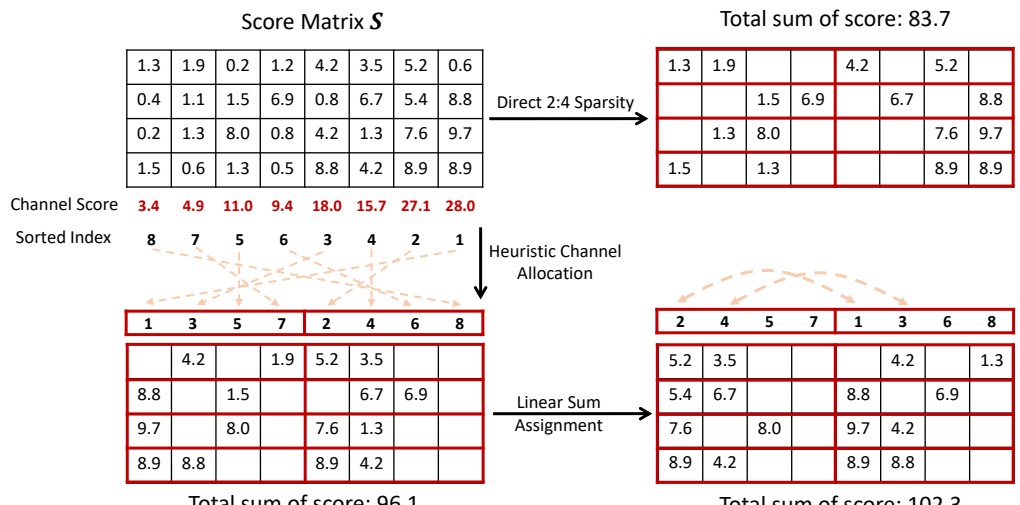

Figure 3: Illustration of Channel Permutation. Given a score matrix **S** assigned by various criteria, directly processing it with 2:4 sparse results in a total sum of the retained score being 83.7. However, by using channel permutation, we could get a final total sum of score 102.3.

existing solutions that directly convert to the N:M format, we propose channel permutation, a new approach that better leverages the pruning metrics by efficiently permuting the weight channels. Channel permutation can work seamlessly with the RIA metric in Section 3 and other prevalent methods such as (Frantar & Alistarh, 2023; Sun et al., 2023).

## 4.1 N:M SEMI-STRUCTURED PRUNING: FORMULATION

We begin by revisiting unstructured PTP in Equation 1. Without loss of generality, we denote the weight importance score as $\mathbf{S} \in \mathbb{R}^{r \times c}$, which can be either RIA or other pruning metrics. It is an NP-hard problem (Frantar & Alistarh, 2023) to find the optimal $\mathbf{M} \in \{0, 1\}^{r \times c}$ in Equation 1. A common surrogate is to maximize the sum of retained weight importance scores as follows:

$$\arg \max_{\mathbf{M}} \sum \mathbf{M} \odot \mathbf{S}, \quad \text{s.t.} \quad ||\mathbf{M}||_0 \leq k. \tag{4}$$

Similarly, for N:M sparsity, every N out of M contiguous elements is zero along each output channel, and the objective is re-formulated as:

$$\arg \max_{\mathbf{M}} \sum_{i=0}^{r} \sum_{k=0}^{\frac{c}{m}} \sum (\mathbf{M} \odot \mathbf{S})_{i,km:(k+1)m}, \quad \text{s.t.} \quad ||\mathbf{M}_{i,km:(k+1)m}||_0 \leq m - n. \tag{5}$$

A simple solution to Equation 5 in existing works (Frantar & Alistarh, 2023; Sun et al., 2023) is directly setting the mask $\mathbf{M}_{i,km:(k+1)m}$ of top N elements in $\mathbf{S}_{i,km:(k+1)m}$ to 1, and 0 otherwise. Nonetheless, this usually leads to sub-optimal solutions. As illustrated in Figure 3, some input channels with similar scores, either large or small, might get stuck in the same weight block $\mathbf{W}_{km:(k+1)m,*}$. Consequently, some large weights might be pruned by mistake, while small ones might be preserved.

## 4.2 CHANNEL PERMUTATION FOR IMPROVED N:M SPARSITY

To address the aforementioned challenge, we present a new channel permutation (**CP**) approach that yields better N:M structures. Note that permuting the input channels of weight can lead to different importance scores **S**. We thus introduce an additional column permutation matrix **P** for **S**, such that the sum of retained weight importance can be further maximized:

$$\arg\max_{\mathbf{M},\mathbf{P}} \sum_{i=0}^{r} \sum_{k=0}^{\frac{c}{m}} \sum (\mathbf{M} \odot (\mathbf{S}\,\mathbf{P}))_{i,km:(k+1)m}, \quad \text{s.t.} \quad \|\mathbf{M}_{i,km:(k+1)m}\|_0 \leq m - n. \tag{6}$$

For ease of presentation in what follows, we denote the *block* of the weight matrix as $\mathbf{W}_{*,km:(k+1)m} \in \mathbb{R}^{r \times m}$, where $k \in \{1, ..., K\}$ and $K$ is the number of blocks. N:M pruning thus occurs row-wise within each block, e.g., only $n$ values are preserved out of $m$ elements for $\mathbf{W}_{i,km:(k+1)m}$. Based on the notation, the proposed channel permutation mainly includes the following two steps.

**Step 1: Heuristic Channel Allocation.** We first calculate the sum of weight importance for each input channel. These channels are then sorted and allocated into $K$ blocks. To maximally retain the important channels in each block with N:M sparsity, we use a heuristic allocation strategy. Given $K$ blocks, we alternately allocate every top-$K$ input channel into each block, and this process is repeated for $m$ times until all channels are allocated. An example is illustrated in Figure 3. Given 8 input channels and 4 output channels, there are 2 blocks under 2:4 sparsity. The top-1 and top-2 channels are allocated to block 1 and block 2, respectively, and similarly to the remaining input channels. Given such a heuristic channel allocation strategy, it can be found that there is a significant enhancement in the sum of retained weight importance scores compared to direct N:M pruning.

**Step 2: Linear Sum Assignment.** Next, we show the heuristic allocation strategy can be further refined. The refining process can be formulated as a linear sum assignment (LSA) problem, which can be efficiently solved by the Hungarian algorithm (Kuhn, 1955). To see this, we can take out one allocated channel from each block; thus, there are $K$ channels to be reassigned to $K$ blocks. It is thus a traditional linear sum assignment problem to find a better one-by-one matching between the $K$ channels and $K$ blocks, such that the weight importance sum in Equation 6 can be further improved. From Figure 3, LSA further improves the score sum by 6.2, with the top-1 and top-2, top-3 and top-4 channels swapped from the heuristic channel allocation.

**Remarks.** Note that the permutation of weight matrices does not affect the output of LLMs. For dense layers, the input activations need to be simultaneously permuted, which can be achieved by permuting the output channels of the previous layer. The exception lies in the residual connection, which can be done with an efficient permutation operator. More details on implementing channel permutation and the Hungarian algorithm are listed in F.

## 5 EXPERIMENTS

### 5.1 SETUP

We evaluate the proposed approach on three popular LLMs: LLaMA 7B-65B (Touvron et al., 2023a), LLaMA2 7B-70B (Touvron et al., 2023b), and OPT 1.3B (Zhang et al., 2022a). We use the public checkpoints of the involved models in the HuggingFace Transformers library [1]. We utilize 3 NVIDIA A100 GPUs, each equipped with 80GB memory. For each model under consideration, we apply uniform pruning to all linear layers, with the exception of embeddings and the head. Specifically, each self-attention module has four linear layers, while each MLP module contains three linear layers for LLaMA model families and two for OPT. All the evaluations are conducted with the same code to make sure the comparison is fair. The detailed settings of tasks, metrics, baseline methods, and calibration data can be found in Appendix B.

### 5.2 UNSTRUCTURED PRUNING

**Main Results.** As highlighted in Table 1, RIA consistently outperforms Wanda and SparseGPT across all scenarios. Notably, our method achieves a 50% improvement in preventing a performance drop of the dense model in comparison to SparseGPT (16% in LLaMA and LLaMA2 model family), and a 17% improvement in preventing a performance drop in comparison to Wanda (13% in LLaMA and LLaMA2 model family). It is essential to note that as the model size increases, the performance gap between models pruned by RIA and the original dense models diminishes significantly.

---

[1]https://huggingface.co/meta-llama, https://huggingface.co/facebook

Table 1: Perplexity results on Wikitext2. We produce the one-shot Post-Training pruning methods with 50% unstructured sparsity on LLaMA, LLaMA2, and OPT models.

| Method | LLaMA 7b | LLaMA 13b | LLaMA 30b | LLaMA 65b | LLaMA2 7b | LLaMA2 13b | LLaMA2 70b | OPT 1.3b |
|---|---|---|---|---|---|---|---|---|
| Dense | 5.68 | 5.09 | 4.77 | 3.56 | 5.47 | 4.88 | 3.32 | 14.62 |
| Magnitude | 17.28 | 20.22 | 7.54 | 5.90 | 16.02 | 6.83 | 5.36 | 1712 |
| Wanda | 7.26 | 6.15 | 5.24 | 4.57 | 6.92 | 5.99 | 4.22 | 18.41 |
| SparseGPT | 7.24 | 6.20 | 5.32 | 4.57 | 6.99 | 6.10 | 4.25 | 27.00 |
| RIA (Ours) | **7.12** | **6.08** | **5.08** | **4.38** | **6.81** | **5.83** | **4.11** | **18.08** |

**Ablation Studies.** To thoroughly evaluate the influence of each element within our RIA equation, we undertook an ablation test, with the outcomes presented in Table 2. We disassembled our formula into several distinct components for closer scrutiny: $||\mathbf{X}||_2$: $\ell_2$-norm of activations; $|\mathbf{W}|$: weight magnitude; $|\mathbf{W}|_{in}$: weight magnitude normalized by the input channels; $|\mathbf{W}|_{out}$: weight magnitude normalized by the output channels; **RI**: stands for relative importance which is the combination of $|\mathbf{W}|_{in}$ and $|\mathbf{W}|_{out}$; and **RIA** ($a = 1.0$) and **RIA** ($a = 0.5$) combines relative importance with input activations with different values of $a$. The selection of $a$ is shown in Figure 9.

Table 2: Ablation Studies of RIA on LLaMA-13B and LLaMA-30B models.

| | LLaMA-13B | LLaMA-30B |
|---|---|---|
| $||\mathbf{X}||_2$ | 9056 | nan |
| $|\mathbf{W}|$ | 20.22 | 7.55 |
| $|\mathbf{W}|_{in}$ | 11.97 | 6.73 |
| $|\mathbf{W}|_{out}$ | 7.80 | 5.55 |
| **RI** | 6.57 | 5.27 |
| **RIA** ($a = 1.0$) | 6.14 | 5.13 |
| **RIA** ($a = 0.5$) | 6.08 | 5.08 |

As illustrated in Table 2, normalizing the weight magnitude through either input or output channels offers a substantial performance boost over merely considering weight magnitude. Interestingly, utilizing Relative Importance alone can match or, in instances like LLaMA-30B, even outperform SparseGPT. A distinguishing feature of Relative Importance is its reliance solely on weight information, eliminating the need for calibration data. In contrast, both Wanda and SparseGPT necessitate calibration data to derive input activations or Hessian matrices. The table also showcases enhancements brought about by the other equation components of RIA.

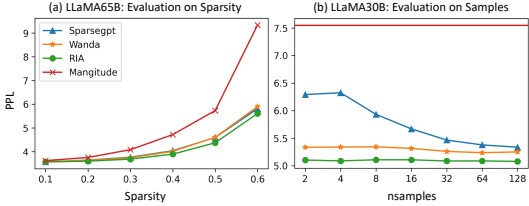

Figure 4: Sensitivity Evaluation on Sparsity and number of calibration samples (nsamples).

**Sparsity.** In Figure 4(a), we examine the effects of varying sparsity levels, ranging from 0.1 to 0.6, on the performance of the LLaMA 65b model. The PPL curves clearly demonstrate that Magnitude pruning is particularly sensitive to increased sparsity levels, with a 60% sparsity level resulting in significant model degradation. In contrast, the SparseGPT, Wanda, and RIA models exhibit more robust performance across all tested sparsity levels, with RIA (green) consistently outperforming the others at every level of sparsity.

**Calibration data.** SparseGPT, Wanda, and RIA all require calibration data for obtaining either input activations or the Hessian matrix. To assess the robustness of our algorithm with respect to the calibration data, we conduct a sensitivity test involving variations in the type of calibration datasets and the number of calibration samples. The influence of calibration datasets is presented in Table 7. Our aim here is to assess the impact of the number of calibration samples.

As illustrated in Figure 4(b), an example of the LLaMA30B model, SparseGPT appears to rely on a larger number of calibration samples, while both Wanda and RIA demonstrate robust performance across varying sample sizes. Notably, RIA consistently outperforms the other methods in all cases.

Table 3: LLaMA2-70B: Zero-Shot Performance of the model with unstructured 50% sparsity compared to the dense model. Bold values denote the best performance across all the post-training pruning methods. An asterisk ("*") signifies performance surpassing that of the dense method.

| Method | Hellaswag | BoolQ | ARC-C | MNLI | RTE | AVG |
|---|---|---|---|---|---|---|
| Dense | 64.77 | 83.70 | 54.44 | 45.81 | 67.87 | 63.32 |
| Magnitude | 60.58 | 71.10 | 49.32 | 32.80 | 60.65 | 54.89 |
| Wanda | 62.70 | 83.27 | 52.50 | **43.19** | 70.84* | 62.50 |
| SparseGPT | 62.36 | 84.26* | **53.07** | 40.29 | 70.76* | 62.15 |
| RIA | **63.22** | **84.77*** | 52.56 | 42.80 | **71.48*** | **62.97** |

Table 4: Unstructured sparsity and semi-structured sparsity results on Wikitext2. We highlight the best performance among all methods within the same sparsity pattern in bold.

| | Method | Unstructured 50% | 2:4 | 2:4+CP w/o LSA | 2:4+CP | 4:8 | 4:8+CP |
|---|---|---|---|---|---|---|---|
| LLaMA2-13b (Dense 4.88) | Magnitude | 6.83 | 8.74 | 8.89 | 8.87 | 7.32 | 7.16 |
| | Wanda | 5.99 | 9.00 | 8.74 | 8.45 | 7.00 | 6.83 |
| | SparseGPT | 6.10 | 8.77 | 8.61 | 8.48 | 7.01 | 6.80 |
| | RIA (Ours) | **5.83** | **8.41** | **8.03** | **7.77** | **6.74** | **6.53** |
| LLaMA2-70b (Dense 3.32) | Magnitude | 5.36 | 6.76 | 6.77 | 6.71 | 5.89 | 5.91 |
| | Wanda | 4.23 | 5.48 | 5.29 | 5.23 | 4.77 | 4.63 |
| | SparseGPT | 4.25 | 5.68 | 5.37 | 5.31 | 4.91 | 4.79 |
| | RIA (Ours) | **4.11** | **5.36** | **5.18** | **5.11** | **4.68** | **4.54** |

**Zero-shot performance.** In Table 3, we present the zero-shot performance of unstructured 50% sparsity of the models pruned with magnitude, Wanda, SparseGPT, and RIA on LLaMA2-70b model. In the last column, we report the average performance across these datasets. As shown in the table, RIA achieves the best performance on 3/5 datasets and also achieves the best average performance across the 5 datasets.

## 5.3 N:M Semi-Structured Pruning

While RIA aims to explore the upper bounds of performance achievable through one-shot PTP methods, combining it with the N:M constraint seeks to realize the actual inference speed by aligning with the present GPU hardware environment. In this subsection, we assess how Channel Permutation (CP) can enhance the performance of one-shot PTP when incorporated with the N:M constraint.

**Main Results.** In this comparison, we assess the performance of unstructured 50% sparsity, 2:4 constraint sparsity, and 4:8 constraint sparsity for Magnitude, Wanda, SparseGPT, and RIA. Additionally, we provide the performance results when applying Channel Permutation (CP) to each of these methods. Directly using step 1 of CP (CP w/o LSA) is also displayed in the table, serving as an ablation test in comparison to the complete one. We present the Perplexity of each pruned model on the Wikitext2 dataset, maintaining the same settings as in Section 5.1.

As presented in Table 4, RIA consistently delivers superior performance across all semi-structured sparsity patterns when employing one-shot PTP. Importantly, when utilizing merely heuristic channel reallocation, every method—with the exception of Magnitude—already exhibits a significant performance improvement. With the incorporation of LSA, the performance is further improved. This highlights our motivation to group the input channels based on their sorted indices, ensuring that similar scaling channels don't end up in the same block.

**Zero-shot Performance.** In Table 6, we present the zero-shot performance of the N:M constraint models pruned using RIA and Wanda. The table's last column also provides the average performance across these datasets. As the table indicates, while there's a performance decline on the Hellaswag and ARC-C datasets, the case of RIA (2:4+CP) performs even surpasses the dense model on BoolQ,

Table 5: LLaMA2-13B: Inference time of different sparsity patterns. Batch size of input sequences is 8 and the sequence length is 128.

| Method | Q/K/V/Out | Up/Gate | Down | Overall |
|---|---|---|---|---|
| unstructured 50% | 0.98× | 0.98× | 0.97× | 0.98× |
| 2:4 (cuTLASS) | 1.21× | 1.23× | 1.23× | 1.22× |
| 2:4 (cuSPARSELT) | 1.64× | 1.65× | 1.62× | 1.63× |

MNLI, and RTE datasets. This highlights the observation that large language models can be prone to overfitting, resulting in an abundance of redundant, unnecessary, or potentially harmful elements. Notably, with the incorporation of CP, there's a remarkable improvement in performance for both Wanda and RIA.

### 5.4 RUNNING TIME ANALYSIS

**Pruning and permutation running time.** We present the actual running time of each algorithm on the largest model involved in this article, LLaMA2-70b. The relative complexity analysis can be found in Appendix G.

- **Pruning time.** We test each algorithm with 128 calibration data. For SparseGPT, the pruning time amounts to 5756 seconds (approximately 1.5 hours). In contrast, Wanda and RIA demonstrate significantly reduced runtime, with times of approximately 611 seconds (approximately 10 minutes) and 627 seconds (approximately 10 minutes, including the execution of calibration samples).

- **Channel permutation time.** For a comparative analysis of execution duration, we present the processing time of a single matrix constructed using our algorithm for the N:M sparsity. This is compared against the greedy method and its variants, which employ escaping strategies to circumvent getting trapped in local minima, as discussed in (Pool & Yu, 2021). The results for different dimensions are provided in Table 11.

**Inference acceleration.** We assess the inference acceleration offered by sparsity in Large Language Models (LLMs). Like SparseGPT (Frantar & Alistarh, 2023), we present data for both the unstructured sparsity and 2:4 sparsity acceleration on GPU relative to the dense model across various components. Theoretically, employing an N:M sparsity can yield up to a 2× speedup when contrasted with dense models. We conduct tests on the Nvidia Tesla A100, utilizing the cuTLASS and cuSPARSELt library for Sparse Matrix-Matrix Multiplication (SpMM) (Mishra et al., 2021) with N:M sparse matrices. These two libraries have been incorporated into the latest release of PyTorch. For unstructured sparsity, we treat it as if it is a dense matrix and assess the actual inference speed. Comprehensive acceleration metrics for each module are outlined in Table 5, showing an acceleration of all the linear layers to be about 1.2× for cuTLASS and 1.6× for cuSPARSELT.

## 6 CONCLUSION

In this article, we have introduced two novel methods, RIA and Channel Permutation, that together establish an effective plug-and-play pipeline for post-training pruning and inference acceleration of large language models. RIA incorporates relative importance and the feature of input activations that create a criterion for pruning the weights of LLMs. Through extensive experiments on prominent LLMs like LLaMA, LLaMA2, and OPT across varying model sizes, we have demonstrated that RIA consistently outperforms existing SOTA one-shot pruning techniques SparseGPT and Wanda, setting a new benchmark for post-training pruning performance. Furthermore, Channel Permutation successfully reduces the performance drop when adapting the model to the N:M constraint by reframing the input channel permutation problem as a combinatorial optimization task and solving it efficiently with the Hungarian algorithm. RIA and Channel Permutation form a seamless "plug-and-play" method, enabling effective one-shot post-training pruning for all current large language models. Furthermore, this method is hardware-friendly, ensuring enhanced inference acceleration.

## ACKNOWLEDGE

This work is supported by the Zhou Yahui Chair professorship of Tsinghua University, the starting funding of the Tsinghua Laboratory of Brain and Intelligence, and the National High-level Talent Program of the Ministry of Science and Technology of China.

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

Table 6: LLaMA2-70B: Zero-Shot Performance of N:M constraint model comparing to the dense model. Bold values denote the best performance across all N:M constraint models. An asterisk ("*") signifies performance surpassing that of the dense method.

| Method | Hellaswag | BoolQ | ARC-C | MNLI | RTE | AVG |
|---|---|---|---|---|---|---|
| Dense | 64.77 | 83.70 | 54.44 | 45.81 | 67.87 | 63.32 |
| Wanda (2:4) | 57.35 | 81.44 | 46.01 | 37.69 | 68.59* | 58.22 |
| Wanda (2:4+CP) | 59.37 | 84.50* | 48.55 | 43.09 | 66.43 | 60.39 |
| Wanda (4:8+CP) | **60.86** | 82.73 | 49.94 | 40.15 | 67.87 | 60.51 |
| RIA (2:4) | 57.13 | 82.78 | 46.76 | 37.39 | 69.31* | 58.68 |
| RIA (2:4+CP) | 58.48 | **85.14*** | 49.15 | **49.08*** | 68.95* | 62.16 |
| RIA (4:8+CP) | 60.44 | 83.58 | **50.43** | 48.69* | **70.04*** | **62.64** |

## A  MOTIVATION OF CHANNEL PERMUTATION

Based on the results presented in Section 5, it is evident that after applying channel permutation, the performance of N:M constraint models significantly improves in comparison to executing N:M sparsity directly. This improvement can be demonstrated through empirical observations made within Large Language Models (LLMs). As shown in Figure 5, pruning with 50% unstructured sparsity versus direct N:M con-

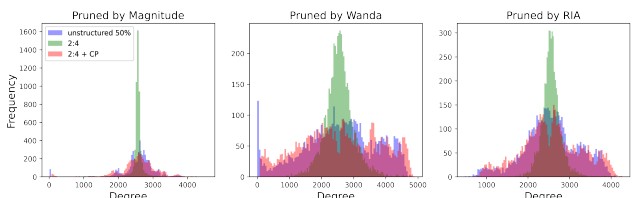

Figure 5: The degree distribution for input channels after pruning.

straint sparsity (2:4 in the figure) produces distinct variances in the distribution of input channels' retained degree. The N:M constraint, by its nature, results in a more uniform pruning pattern. With an example from the first row of Figure 3, when directly pruning with 2:4 constraint sparsity, if the input channels with universally larger scores get stuck in the same block, some of the weights will be "wrongly" pruned. Conversely, the block with universally smaller scaling scores will "wrongly" preserve some weights. One diminishes while the other grows, resulting in a more uniform pruning, a smaller variance of the retained degree distribution of input channels, and therefore, a lower total sum of scores after pruning. In contrast, unstructured pruning takes on a broader perspective, pruning weights within their global context. This expansive methodology inherently introduces more variability, leading to a larger variance in the retained degree of input channels after pruning. The channel permutation method proposed in this article tackles this challenge smoothly by distributing similar scaling input channels into different blocks, allowing weights that should be retained to be preserved as much as possible. The post-channel-permutation distribution (2:4 + CP) of input channels' retained degrees can be seen in Figure 5, closely mirroring the results of unstructured pruning.

## B  DETAILED EXPERIMENTAL SETTINGS

**Tasks and Metrics.** We mainly evaluate language modeling and zero-shot classification. Our initial assessment of language modeling involves a comprehensive evaluation of perplexity (PPL), where lower values indicate better performance. This evaluation is conducted on the test set of Wikitext2 (Merity et al., 2016). For further evaluation, we also conduct experiments on zero-shot classification to assess the ability of the sparse model to correctly classify objects or data points into categories it has not seen during training across five common-sense datasets: Hellaswag, BoolQ, ARC-Challenge, MNLI, and RTE, compared to the performance of the dense model. We run the experiments with the public GitHub benchmark EleutherAI/lm-evaluation-harness (Gao et al., 2021).

Table 7: Assessing across various calibration and evaluation datasets. (LLaMA2-13B)

| Eval. dataset | PTB | | | wikitext2 | | | c4 | | |
|---|---|---|---|---|---|---|---|---|---|
| Calib. dataset | wikitext2 | c4 | PTB | wikitext2 | c4 | PTB | wikitext2 | c4 | PTB |
| Magnitude | | 146.35 | | | 6.83 | | | 9.38 | |
| Wanda | 68.49 | 69.70 | 63.58 | 5.85 | 5.97 | 5.89 | 8.43 | 8.30 | 8.25 |
| SparseGPT | 72.94 | 72.31 | **59.10** | **5.69** | 6.03 | 5.90 | 8.50 | 8.22 | 8.30 |
| RIA | **67.58** | **68.69** | 67.88 | 5.75 | **5.83** | **5.83** | **8.07** | **8.03** | **8.08** |

**Baselines.** For unstructured pruning, we compare with: 1) magnitude pruning (Zhu & Gupta, 2017), the most prevalent pruning approach; and two more recent state-of-the-art works on LLM pruning: 2) SparseGPT (Frantar & Alistarh, 2023) and 3) Wanda (Sun et al., 2023). These methods can be evaluated both on unstructured pruning and N:M semi-structured pruning in the following sections.

**Calibration Data.** We employ 128 samples from the C4 dataset (Raffel et al., 2019) for all models, and each sample contains 2048 tokens. This also aligns with the settings in baseline methods for a fair comparison. Note that we also discuss the choice of calibration data across different datasets, and more details can be found in Appendix C.

## C    SENSITIVITY TEST ON CALIBRATION DATASETS

Figure 2 illustrates that different datasets exhibit varying data distributions, which in turn impact the input activations. Consequently, we conduct experiments employing different calibration datasets to evaluate the robustness of the pruned model. Specifically, we utilized three different datasets—Wikitext2, C4, and PTB (Marcus et al., 1993)—with 128 samples, each containing 2048 tokens, for calibration purposes, and evaluated the performance on Wikitext2. RIA is compared with Wanda and SparseGPT, with the results summarized in Table 7. RIA surpasses other algorithms in the majority of scenarios, except when using identical calibration and evaluation datasets for PTB and wikitext2. This exception can be primarily attributed to the weight reconstruction process. A more congruent distribution tends to yield enhanced performance through reconstruction. Therefore, we explore integrating the reconstruction approach with Wanda and RIA in Appendix D. Furthermore, looking at the results across different calibration datasets, the results of RIA are more stable, which indicates that RIA is more robust than the calibration data.

## D    WEIGHT RECONSTRUCTION

Following the approach of OBS (Hassibi et al., 1993), which replaces the fine-tuning process with weight reconstruction using calibration data, this method has gained traction in LLMs post-pruning as an alternative to fine-tuning and retraining. SparseGPT (Frantar & Alistarh, 2023) extends this idea by permitting partial updates, thereby reducing computational complexity. The specific formula for this is detailed in (Frantar & Alistarh, 2023). This weight reconstruction technique is essentially a tool embraced by all PTP methods that revise the objective function of equation 1 to 7.

$$\arg \min_{\mathbf{M}_l, \hat{\mathbf{W}}_l} ||\mathbf{W}_l \cdot \mathbf{X}_l - (\mathbf{M}_l \odot \hat{\mathbf{W}}_l) \cdot \mathbf{X}_l||_2^2 \tag{7}$$

In this formulation, $\hat{\mathbf{W}}_l$ represents the weight matrix after undergoing the reconstruction process. In this section, we evaluate Wanda and RIA based on their adoption of weight reconstruction and juxtapose their performance with that of SparseGPT. Our findings are presented in Table 8. We omit the results from PTB as their PPLs are excessively high, rendering them of no reference value. It's evident that RIA+rec outperforms other reconstruction-based algorithms. However, except when Wikitext2 is used both for calibration and evaluation, the reconstruction does not appear to enhance performance. Therefore, we have chosen not to employ this weight reconstruction approach in our main text.

Table 8: Assessment of Post-Training Pruning Methods Integrating Weight Reconstruction (LLaMA13B).

|  | wikitext2 | | | c4 | | |
|---|---|---|---|---|---|---|
|  | wikitext2 | c4 | PTB | wikitext2 | c4 | PTB |
| Magnitude | 6.38 | | | 9.38 | | |
| Magnitude + rec | 5.84 | 6.07 | 6.17 | 8.48 | 8.30 | 8.62 |
| Wanda | 5.85 | 5.97 | 5.89 | 8.43 | 8.30 | 8.25 |
| Wanda+rec | 5.70 | 6.00 | 5.91 | 8.57 | 8.29 | 8.35 |
| SparseGPT | 5.69 | 6.03 | 5.90 | 8.50 | 8.22 | 8.30 |
| RIA | 5.75 | **5.83** | **5.83** | **8.07** | **8.03** | **8.08** |
| RIA+rec | **5.57** | 5.86 | **5.83** | 8.20 | **8.03** | 8.21 |

## E  CHANNEL CORRUPTIONS AND STRUCTURED PRUNING

**Evidence of Channel Corruption.**  In Section E, we discussed the occurrence of channel corruption during the application of Wanda for pruning LLMs. Figure 6 illustrates the distribution of connections per input channel in the weight matrix following pruning by both Wanda and RIA, where degree denotes the number of connections a node has. Our findings reveal that in certain layers, Wanda results in roughly 10% of the channels experiencing corruption. Conversely, RIA ensures that no channels undergo corruption.

**Detrimental Impact of Channel Corruption on Model Performance.**  To assess the effects of full channel pruning on model performance, we utilize LLM-pruner (Ma et al., 2023), a SOT) method for structured pruning in LLMs. These experiments are designed to demonstrate the influence of channel corruption. Moreover, we adopt a pruning strategy based on activation $||X||$ as a fundamental comparative method, which serves not only as a test of structured pruning but also as an ab-

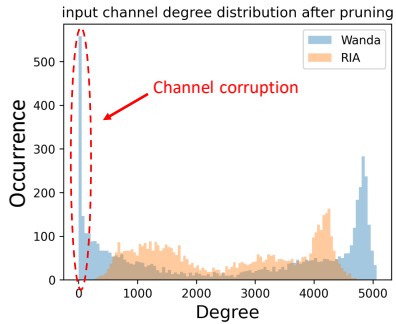

Figure 6: Comparison of Channel Corruption in Wanda and RIA. The plot shows the remaining node degree distribution post-pruning.

lation study for our RIA method. We compare the structured pruning methods with Wanda and RIA with both unstructured sparsity results and semi-structured sparsity results. In LLM-pruner (Ma et al., 2023), they introduce a skip-layer pruning strategy that skips the pruning process for the first several layers and the last few layers. To provide a thorough evaluation, we perform experiments on pruning across all model layers, as elaborated in Table 10.

We have the following observations:

1) The overall performance of different sparsity patterns reveals that structured sparsity is less effective than semi-structured sparsity even at a low sparsity level of 25%, highlighting the detrimental impact of channel corruption on model effectiveness. Across all sparsity degrees and patterns, RIA consistently outperforms Wanda, demonstrating its superior performance.

2) The introduction of channel permutation has been found to boost performance as the model shifts towards semi-structured sparsity. This outcome is consistent with the observations detailed in the main text.

**End-to-End Inference Latency of N:M Sparsity and Structured Sparsity.**  We offer the LLM inference latency with structured sparsity and 2:4 sparsity. Our experiment is on LLaMA2-7b deployed on 2 NVIDIA A100s with 80 GB memory, where the input context sequence length is 128. We increase the batch size and record the inference latency accordingly. According to Table 9,

Table 10: Comparative performance of structured and semi-structured pruning at various sparsity levels: experiments on LLaMA2-7B with calibration on C4 with 128 samples and evaluation on Wikitext2 using PPL metrics. The table includes experiments pruning on all layers.

|  | 1:4 (25% sparsity) | 2:4 (50% sparsity) | 3:4 (75% sparsity) |
|---|---|---|---|
| LLM-Pruner (structured) | 28.88 | nan | 13570 |
| Activation-based (structured) | 39.54 | 10467 | nan |
| Wanda (semi-structured) | 5.94 | 12.15 | 2863.3 |
| RIA (semi-structured) | 5.90 | 11.27 | 1891.13 |
| RIA + CP (semi-structured) | 5.81 | 10.12 | 1532.72 |
| Wanda (unstructured) | 5.68 | 6.92 | 1506.13 |
| RIA (unstructured) | 5.56 | 6.81 | 267.61 |

structured sparsity (50%) marginally outperforms the 2:4 sparsity model as the batch size increases. Importantly, note that with smaller batch sizes, both structured 50% sparsity and 2:4 sparsity models do not significantly boost inference speed. However, as the batch size grows, the acceleration for 2:4 sparsity approximates 1.5x, whereas the structured 50% sparsity tends to reach about 1.7x acceleration.

In conclusion, N:M semi-structured sparsity emerges as the optimal compromise between performance and latency. Although structured pruning offers the additional advantage of acceler-

Table 9: The inference latency of 2:4 sparsity and structured 50% sparsity with varying batch sizes on LLaMA2-7b model.

| Batch Size | 1 | 8 | 16 | 64 |
|---|---|---|---|---|
| dense | 281.13 ms | 547.82 ms | 1011.81ms | 3742.76 ms |
| structured sparsity (50%) | 238.92 ms | 326.14 ms | 616.13 ms | 2181.69 ms |
| 2:4 sparsity | 225.60 ms | 357.48 ms | 731.81 ms | 2495.41 ms |

ation at the same sparsity level, the associated performance decrease is significant, even at very low sparsity levels.

# F    IMPLEMENTATION DETAILS OF CHANNEL PERMUTATION

## F.1    OUTPUT CHANNEL REORDERING

From a computational efficiency perspective, directly implementing permutation on input vectors introduces extra runtime overhead for extracting the permuted index and applying it to the input vectors. However, an alternative approach can be considered. Given that each input serves as the output of the preceding layer, we can permute the output channels of the previous layers' weights to serve as the permuted index for the next layers' input channels. This eliminates the need to permute the input index separately, resulting in time savings.

However, two key considerations must be taken into account:

a) The Q, K, and V projection layers within a single module must be treated as a unified matrix, concatenated into a single entity that shares the same input channel indices. This ensures that the input index permutation remains identical across these layers, avoiding the need for additional permutations. In the case of LLaMA, the MLP module presents a similar scenario, as it contains parallel layers, namely, down_proj and gate_proj, which also need to be concatenated into a single matrix.

b) Another important factor to note is that this strategy cannot be applied when dealing with models featuring residual connections. This is because inputs in such modules also originate from the previous module. As a result, permutations must be applied to the input activations at the beginning of each module. However, this process does not significantly impact execution time. We've modified the RMS normalization layer — which typically follows the Residual connection — using Triton (Tillet et al., 2019). This ensures this layer's output channel indices align with the subsequent layers' input indices. In our tests, the time taken for this adjustment is negligible.

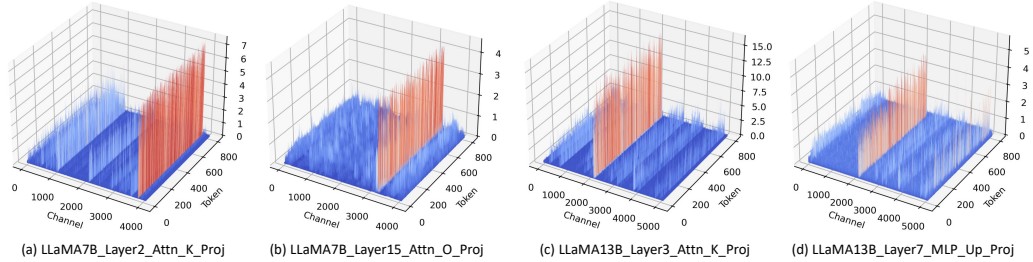

Figure 7: Outliers in LLaMA-7B and LLaMA-13B.

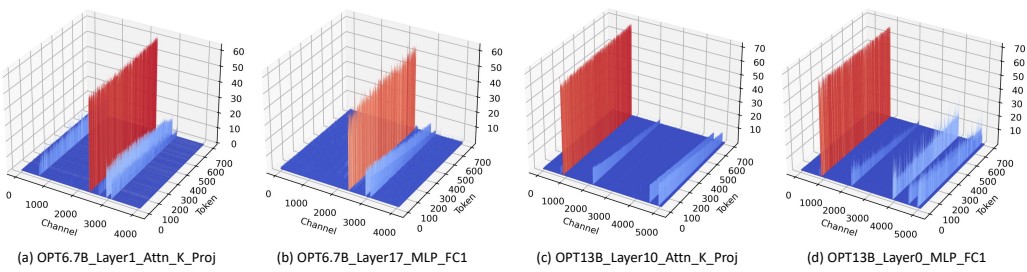

Figure 8: Outliers in OPT-6.7B and OPT-13B.

## F.2 COMMUNITY-BASED PARTITIONING

In optimizing the process of Channel Permutation, we employ a method that significantly accelerates execution by partitioning the input channels into several communities. This strategic division reduces the runtime of channel permutation to $\frac{1}{n^2}$, where $n$ represents the number of communities. The primary source of this reduction in execution time stems from the more efficient computation of the sum of scores following N:M pruning, achieved when input channels are allocated into distinct blocks. The experimental results presented in the main text are grounded on these optimization techniques.

## G COMPLEXITY ANALYSIS OF RIA AND CHANNEL PERMUTATION

**RIA.** We provide a summary of the computational complexity of various Post-Training Pruning (PTP) algorithms. For SparseGPT, the time complexity is approximately $O(d_{\text{hidden}}^3)$ (Frantar & Alistarh, 2023), whereas both RIA and Wanda exhibit similar time complexities of approximately $O(d_{\text{hidden}}^2)$ (Sun et al., 2023).

**Channel Permutation.** The process of channel reallocation to obtain the sorted input channel indices is straightforward. Computation of the score matrix $S$ involves determining the outcome of placing each objection into every box and summing the results while processing the 2:4 constraint within each block. Consequently, the time complexity for this computation is $O\left(\left(\frac{c}{M}\right)^2 \times r \times M\right)$,

Table 11: Permutation time (seconds) for Greedy algorithm and Channel Permutation.

|  | $4096 \times 4096$ | $5120 \times 5120$ | $6656 \times 6656$ | $8192 \times 8192$ |
|---|---|---|---|---|
| Greedy | 252.1 | 495.4 | 818.7 | 1563.6 |
| Greedy + 100 escapes | 845.3 | 1349.1 | 1896.4 | 3592.3 |
| Channel Permutation | 6.2 | 8.1 | 11.5 | 15.3 |

Table 12: Integration of Post-Training Pruning with Quantization Methods.

|  | Magnitude | SparseGPT | Wanda | RIA |
|---|---|---|---|---|
| Unstructured (50% sparsity) | 16.02 | 6.99 | 6.92 | 6.81 |
| Unstructured (50% sparsity) With GPTQ | 15.21 | 7.59 | 7.41 | 7.28 |
| Unstructured (50% sparsity) With AWQ | 17.12 | 7.15 | 7.10 | 6.97 |

where $r \times M$ represents the computational complexity for the N:M constraint within each block. The time complexity of the Hungarian algorithm is $O\left(\left(\frac{c}{M}\right)^3\right)$.

The greedy method (Pool & Yu, 2021) is challenging to implement in LLMs due to its prohibitive running time, as shown in Table 11. We conduct only a single experiment for performance comparison on LLaMA2-13b, which took us 3 days. We tested the greedy method with 100 escape iterations to handle the permutation, the PPL on wikitext2 with a permuted 2:4 constraint is 7.83 which is just comparable to our CP method (7.77). However, the CP's execution time is just about 1 hour, making it possible to be applied in the LLMs.

## H   HUNGARIAN ALGORITHM

Given a bipartite graph with $N$ left vertices and $N$ right vertices, depicted by matrix $\mathbf{S}$, where $\mathbf{S}_{ij}$ represents the weight between the left vertex $i$ and the right vertex $j$, the algorithm aims to identify the ideal matching that minimizes the aggregate weight. This goal is captured in the subsequent objective function:

$$\min \sum_{i=1}^{N} \sum_{j=1}^{N} \mathbf{S}_{ij} \times \mathbf{X}_{ij}. \tag{8}$$

In this Equation, $\mathbf{X}_{ij}$ is a binary determinant that showcases whether the left vertex $i$ is paired with the right vertex $j$. In our scenario, the initial first group of input indices is treated as vertex $i$, with the incomplete boxes acting as vertex $j$. This transition into an LSA problem is seamlessly facilitated by the Hungarian algorithm, which subsequently derives the optimal permutation. After rearranging the first indices of the blocks, we apply the same procedure to the subsequent groups in sequence.

## I   INTEGRATION OF POST-TRAINING PRUNING WITH QUANTIZATION METHODS

Our experiments utilized the LLaMA2-7b model, focusing on two quantization methods: GPTQ (Frantar et al., 2022) and AWQ (Lin et al., 2023). The calibration is performed using the C4 dataset, and evaluations are done on Wikitext2. We present our results in Table 12.

Our finding suggests that this integration can reduce memory and inference acceleration with a slight detriment to performance.

We have identified two primary strategies for merging post-training pruning with quantization: (a) first pruning, then quantization, and (b) first quantization, then pruning. Our finding indicates a preference for (a) pruning before quantization. This is potentially because for (b), conventional block-wise quantization relies merely on the minimum and maximum weights within a block, and pruning can lead to a reduction in these extremes, thereby potentially aiding quantization. In contrast, (a) quantization before pruning adversely affects the computation of weight importance for pruning, resulting in diminished performance.

## J   THE EXPLANATION OF PREVENTING PERFORMANCE DROP FROM DENSE

The primary goal of post-training pruning methods is to minimize the performance drop compared to the dense model. Therefore, designing metrics that offer a fair evaluation of new methods against this baseline (dense performance) is crucial. Consider two post-training pruning methods, A and B, with perplexities P(A) and P(B), respectively. Let P(D) represent the baseline perplexity of the dense

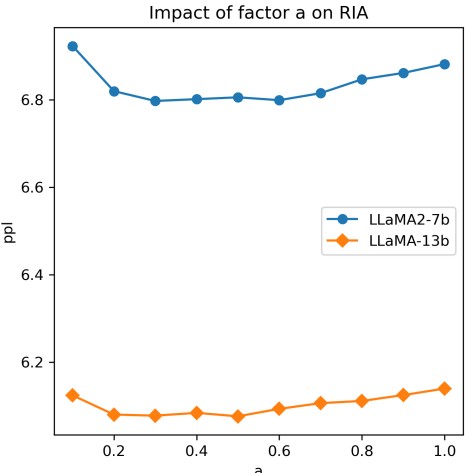

Figure 9: We evaluate how to choose the factor 'a' inside the formula of RIA with LLaMA2-7B and LLaMA-13b model. The calibration dataset is C4 with 128 samples, and the evaluation is on wikitext2. The sparsity is fixed at 50%. From the results, there are no significant differences between a=0.2 to a=0.5. For simplicity, we select a=0.5 in the main text to execute all the evaluations.

network. The effectiveness of method B in preventing performance degradation compared to method A can be quantified as $\frac{P(A)-P(B)}{P(A)-P(D)}$. This formula allows for a fair assessment of the new method relative to previous approaches, taking into account the performance of the dense model in the post-training pruning context. Using this calculation, the RIA method demonstrates a 17% improvement over Wanda and a 50% improvement over SparseGPT in preventing performance drops from the dense model consistently across all tested models.

