# OpenReview forum: "Plug-and-Play: An Efficient Post-training Pruning Method for Large Language Models"
_ICLR.cc/2024/Conference — ICLR 2024 poster_

### Official Review · Reviewer_dzPo · 2023-10-31

**Soundness:** 3 good
**Presentation:** 3 good
**Contribution:** 3 good
**Rating:** 8
**Confidence:** 4

**Summary:**

This paper introduces a post-training N:M pruning solution for LLMs. The method is built upon two components: (1) relative importance and activation, which considers both the weights and activations within LLMs for a better weight importance estimation; and (2) channel permutation, which can better preserve important weights under n:m sparsity through rearranging channel orders. Experiment results demonstrate the effectiveness of the proposed method and its superiority over existing baselines.

**Strengths:**

1. The channel permutation and the Hungarian algorithm are novel techniques, and the experiments demonstrate the effectiveness under n:m sparsity
2. The paper is easy to follow, and the authors conduct extensive experiments.

**Weaknesses:**

1. The first technique, relative importance, and activation, seems to be an incremental improvement.
2. In Section 5.3, which discusses N:M sparsity, I was anticipating experimental results on smaller LLMs such as Llama2-7b, and a higher sparsity ratio than 50%. This would potentially highlight the advantages of the proposed method over SparseGPT and Wanda more effectively. Could the authors provide their insights on this?
3. Regarding the inference latency under n:m sparsity, I would like to suggest that instead of providing layer-wise speedup, the authors could consider providing end-to-end latency for the pruned LLMs. My reason for this suggestion is that I am curious about whether n:m sparsity is indeed an effective structured pruning pattern within LLMs, especially when compared to pure structured pruning methods such as LLM-Pruner. Could the authors provide their perspective on this?

**Questions:**

Please refer to the weaknesses section.

---

> ### Author Response · Authors · 2023-11-21
> **Response to Reviewer 4 (1/2)**
>
> Dear Reviewer hn9p,
>
> Thank you for your insightful feedback and positive evaluation of our manuscript. We appreciate your recognition of the strengths of our work, particularly the novelty of our channel permutation technique and the comprehensiveness of our experimental evaluation. We address your concerns and questions below:
>
> **Weakness 1: The first technique, relative importance, and activation, seems to be an incremental improvement.**
>
> **Reply:** Thank you for your comments. We acknowledge that the improvement of RIA over prior work is modest, around 0.1 - 0.2 points. However, it is important to establish the baseline for this evaluation. If the baseline is zero, an improvement of 0.1-0.2 points may seem negligible. Yet, our objective is to recover the performance of the dense model after pruning. Therefore, the relevant improvement should be measured in relation to the performance drop from the dense model. As reported in our article: "Notably, our method achieves a 50% improvement in mitigating the performance drop of the Dense model compared to SparseGPT (16% in LLaMA and LLaMA2 model family), and a 17% improvement compared to Wanda (13% in LLaMA and LLaMA2 model family)." We believe that any improvement over 10% should not be considered incremental, and this improvement is consistent across all models in our study.
>
> We apologize for not providing a detailed formula to compute this percentage, which may have led to confusion. To address the reviewer's concern, we have added an explanation of this formula in Appendix J of the newly uploaded manuscript. Here is a simplified version for clarity:
>
> Given two post-training pruning methods, A and B, with perplexities P(A) and P(B), and the baseline perplexity P(D) of the dense network, the percentage by which method B prevents the performance drop compared to method A can be computed as $\frac{P(A) - P(B)}{P(A) - P(D)}$. This formula provides a fair evaluation of the newly proposed method compared to previous methods, considering the dense model's performance in the post-training pruning field. Using this calculation, the improvement of RIA in preventing performance drops compared to Wanda is 17% across all models, and 50% compared to SparseGPT across all models.
>
> **Weakness 2:** In Section 5.3, which discusses N:M sparsity, I was anticipating experimental results on smaller LLMs such as Llama2-7b, and a higher sparsity ratio than 50%. This would potentially highlight the advantages of the proposed method over SparseGPT and Wanda more effectively. Could the authors provide their insights on this?
>
> **Reply:**  Thank you for your suggestion. In response, we have conducted additional experiments with varying sparsity levels of N:M sparsity — 25% (1:4 sparsity), 50% (2:4 sparsity), and 75% (3:4 sparsity) — on the Llama2-7b model with the default setting in our article.
>
> To explore this further, we compared two pruning strategies: pruning from layer 4 to layer 32 which is the solution proposed by LLM-pruner to prevent the performance drop and pruning all layers. Our findings confirm that preserving the first and last layers indeed mitigates performance degradation.
>
> Our results indicate that at a 75% sparsity level, there is a significant decline in performance. Although in comparison to Wanda, there is a significant improvement in RIA, the perplexity goes too high to be implemented in real cases.
>
> Here is a summary of our results:
>
> **Pruning from layer 4 to layer 30:**
> |                               | 1:4 (25% sparsity) | 2:4 (50% sparsity) | 3:4 (75% sparsity) |
> | ----------------------------- | ------------------ | ------------------ | ------------------ |
> | LLM-Pruner (structured)       | 26.04              | 77.90              | 464.34             |
> | Activation-based (structured) | 30.23              | 525.5              | 8692.2             |
> | Wanda (semi-structured)       | 5.84               | 9.21               | 432.6              |
> | RIA (semi-structured)         | 5.82               | 8.91               | 442.6              |
> | RIA + CP (semi-structured)    | 5.74               | 8.03               | 436.1              |
>
> **Pruning all the layers:**
>
> |                               | 1:4 (25% sparsity) | 2:4 (50% sparsity) | 3:4 (75% sparsity) |
> | ----------------------------- | ------------------ | ------------------ | ------------------ |
> | LLM-Pruner (structured)       | 28.88              | nan                | 13570              |
> | Activation-based (structured) | 39.54              | 10467              | nan                |
> | Wanda (semi-structured)       | 5.94               | 12.15              | 2863.3             |
> | RIA (semi-structured)         | 5.90               | 11.27              | 1891.13            |
> | RIA + CP (semi-structured)    | 5.81               | 10.12              | 1532.72            |
>
> [1] LLM-Pruner: On the Structural Pruning of Large Language Models, NeurIPS 2023

---

> ### Author Response · Authors · 2023-11-21
> **Response to Reviewer 4 (2/2)**
>
> **Weakness 3:** Regarding the inference latency under n:m sparsity, I would like to suggest that instead of providing layer-wise speedup, the authors could consider providing end-to-end latency for the pruned LLMs. My reason for this suggestion is that I am curious about whether n:m sparsity is indeed an effective structured pruning pattern within LLMs, especially when compared to pure structured pruning methods such as LLM-Pruner. Could the authors provide their perspective on this?
>
> **Reply:** Thank you for your valuable suggestions and questions. We separated your question into two parts:
>
> 1) The performance of N:M sparsity and structured sparsity. We adopted LLM-pruner as the representative structured pruning method to compare with our methods. The results are shown in the above reply. We use the detailed hyperparameter setting: C4 with 128 samples as calibration data, and Wikitext2 as the evaluation data. The PPL of structured pruning (LLM-pruner) increases significantly with just 25% sparsity. This suggests that structured pruning is not effective in post-training pruning scenarios, as the pruned information from the entire channels cannot be recovered without retraining or fine-tuning. Also, To further address the reviewer's concern, we add **Appendix E** to explain this phenomenon in detail.
>
>
> 2. We here offer the inference latency of the structured pruned model with 50% sparsity and 2:4 sparsity model. Our experiment is on LLaMA2-7b, the sequence length of the input is 12. The deployment is on 2 Nvidia A100 80GB in parallel on Ubuntu 22.04.1 system. Here, we offer the inference latency with varying batch sizes.
>
> |                         | 1         | 8         | 16        | 64                 |
> | ----------------------- | --------- | --------- | --------- | ------------------ |
> | dense                   | 281.13 ms | 547.82 ms | 1011.81ms | 3742.76 ms |
> | structured 50% sparsity | 238.92 ms | 326.14 ms | 616.13 ms | 2181.69 ms         |
> | 2:4 sparsity            | 225.60 ms | 357.48 ms | 731.81 ms | 2495.41 ms         |
>
> The table shows that structural 50% sparsity marginally outperforms the 2:4 sparsity model as batch sizes increase. It's important to note that when dealing with smaller input batches, both structured 50% sparsity and 2:4 sparsity models do not significantly boost inference speed. However, as the batch size grows, the acceleration for 2:4 sparsity approximates 1.5x, whereas the structured 50% sparsity tends to reach about 1.7x acceleration.
>
> In conclusion, given the performance and the inference latency analysis, although structured pruning could provide a faster acceleration than N:M sparsity with the same sparsity level, the performance drop is unignorable even when the sparsity is very low. Therefore, in the current situation, the N:M constraint sparsity is the only sparsity pattern that strikes the balance between performance and acceleration speed.
>
> We thank the suggestion the Reviewer gave and we added this part in Appendix E to offer a comprehensive comparison of N:M sparsity and structured sparsity.

---

> > ### Comment · Reviewer_dzPo · 2023-11-22
> > **Response to rebuttal**
> >
> > Thanks for the response. It addresses most of my concerns. I will improve the rating score.

---

> ### Author Response · Authors · 2023-11-22
> **Thanks for increasing your score**
>
> Dear Reviewer,
>
> We are happy to hear that your concerns have been addressed. Thanks for your support!
>
>
> Best wishes,
>
> Authors

---

### Official Review · Reviewer_hn9p · 2023-10-31

**Soundness:** 3 good
**Presentation:** 3 good
**Contribution:** 2 fair
**Rating:** 6
**Confidence:** 4

**Summary:**

* The paper introduces a relative importance pruning metric which leads to more uniform pruning patterns.
* The paper proposes to apply channel permutations found by a scalable heuristic in order to relax the pattern restrictions imposed by n:m pruning.
* Both techniques are evaluated on Llama models for perplexity and zero-shot tasks.

**Strengths:**

* The proposed techniques are relatively simple to implement in practice and in particular the channel reordering seems to be quite effective.
* The paper is easy to understand and provides clear visualizations of the key algorithms.
* Evaluation is carried out on strong Llama models and not just on older OPT ones.
* The Appendix contains interesting additional studies like combining RIA with SparseGPT reconstruction.
* The paper also considers practical acceleration of sparse models in Table 5.

**Weaknesses:**

* The observation that encouraging a more uniform sparsity pattern is beneficial was also made by Wanda, RIA seems to be an extension of that (also across columns). Similarly, that permutation reordering can be helpful for n:m sparsity was found by [Pool & Yu, 2021], this paper only introduces a simpler but more scalable heuristic for finding such a permutation, based on average activation values. While there is some novelty, it is not particularly high.
* For unstructured sparsity, the improvements of RIA over prior work are relatively small at around 0.1-0.2 points in perplexity. The impact of the more advanced linear sum assignment permutation method also seems rather minor. At the same time, perplexity increases from the dense baseline are still quite large, especially for 2:4. Hence, it is not clear how useful the corresponding sparse models would be in practice.
* There does not appear to be any code in the Supplementary. I hope the authors plan to open-source their work to aid reproducability.

While I do not think that the paper brings particularly strong new ideas or practical results, I find it interesting that encouraging even more uniformity than Wanda is beneficial, and that permutation reordering is quite effective for n:m pruning even for massive LLMs. Hence, I am currently leaning towards acceptance.

**Questions:**

* How does 4:8 perform with channel permutations in the setup of Table 3?
* Did you carry out any additional ablation studies around parameter a other than Table 2? I am curious if a = 0.5 is generally the sweet spot or if it was just picked for simplicity.

---

> ### Author Response · Authors · 2023-11-21
> **Response to Reviewer 3 (part 1/2)**
>
> Dear Reviewer hn9p,
>
>
>
> Thank you for your thoughtful evaluation and constructive feedback on our manuscript. We appreciate your recognition of the strengths of our work, including its simplicity, clarity, and practicality. Your insights have been instrumental in guiding our revisions and enhancements. We address your concerns and questions as follows:
>
>
>
> **Weakness 1:** The observation that encouraging a more uniform sparsity pattern is beneficial was also made by Wanda, RIA seems to be an extension of that (also across columns). Similarly, that permutation reordering can be helpful for n:m sparsity was found by [1], this paper only introduces a simpler but more scalable heuristic for finding such a permutation, based on average activation values. While there is some novelty, it is not particularly high.
>
>
>
> **Reply:** We are grateful for your insightful feedback and the opportunity to clarify the unique contributions of our work. Our approach distinguishes itself in two critical aspects.
>
> Firstly, RIA incorporates a consideration of the relative importance of weights. As written in the article, "In practice, we find similar issues also exist in other prevalent pruning metrics, e.g., Wanda prunes around 600 channels out of 5120 channels, with more than 10% channels corrupted. Given that well-trained LLMs contain unique information in the input and output channels, it is critical to avoid channel corruption in post-training pruning." The proposal of relative importance adopts a more uniformly pruning structure that prevents this issue. To address the reviewer's concern, we created Appendix E in the newly uploaded manuscript that explains well the motivation to introduce the relative importance. The performance across all the tests involved in this article, for instance, PPL across all the models on both unstructured and semi-structured sparsity, the sensitivity test of sparsity and number of samples, zero-shot performance, suggests that incorporating relative importance into the pruning metrics offers a consistent advantage against Wanda.
>
> Secondly, the proposal of the channel permutation in this article is to offer a solution to introduce N:M sparsity in large language models. As we introduced in Appendix G: "The greedy method [1] is challenging to implement in LLMs due to its prohibitive running time, as shown in Table 1. We conducted only a single experiment for performance comparison on LLaMA2-13b, which took us 3 days. We tested the greedy method with 100 escape iterations to handle the permutation, the PPL on Wikitext-2 with a permuted 2:4 constraint is 8.01, which is comparable to our CP method (7.99). However, the CP's execution time was just 30 minutes, making it possible to be applied in LLMs." This is a big improvement since no previous works of post-training pruning really stepped into the N:M sparsity field because of the big drop in performance. However, in this article, with an acceptable channel permutation time, the case of RIA (2:4+CP) performs and surpasses the dense model with a large amplitude on BoolQ, MNLI, and RTE datasets of zero-shot performance. We want to draw more attention to the application of how to utilize the N:M constraint since it is currently the best way to retain performance while also offering inference speedup in real cases because it is hardware-friendly. Our method offers a solution to make N:M sparsity possible to be adopted in deploying LLMs.
>
> In addition, as commented in the Reviewer's content, it is true that separating the channels into groups is heuristic, but it has its nature. To better explain our motivation for introducing this and to address the reviewer's concern, we added context in Appendix A that shows the reason why our method can have better results. In short, the post-channel-permutation distribution (2:4 + CP) of input channels’ retained degrees can be seen in Figure 5, closely mirroring the results of unstructured pruning which is the global optimal of the specific pruning metrics (for instance RIA, Sparsegpt, and RIA).
>
> In summary, while our work builds on existing concepts in the field, the refinements and innovations we introduce, particularly in terms of scalability and efficiency, contribute significantly to the practical application of N:M sparsity in large model training.
>
> [1] Channel permutations for N: M sparsity, NeurIPS 2021

---

> > ### Author Response · Authors · 2023-11-21
> > **Response to Reviewer 3 (part 2/2)**
> >
> > **Weakness 2:** Reviewer Comment: For unstructured sparsity, the improvements of RIA over prior work are relatively small at around 0.1-0.2 points in perplexity. The impact of the more advanced linear sum assignment permutation method also seems rather minor.
> >
> > **Reply:** Thank you for your comments. We acknowledge that the improvement of RIA over prior work is modest, around 0.1 - 0.2 points. However, it is important to establish the baseline for this evaluation. If the baseline is zero, an improvement of 0.1-0.2 points may seem negligible. Yet, our objective is to recover the performance of the dense model after pruning. Therefore, the relevant improvement should be measured in relation to the performance drop from the dense model. As reported in our article: "Notably, our method achieves a 50% improvement in mitigating the performance drop of the Dense model compared to SparseGPT (16% in LLaMA and LLaMA2 model family), and a 17% improvement compared to Wanda (13% in LLaMA and LLaMA2 model family)." We believe that any improvement over 10% should not be considered insignificant, and this improvement is consistent across all models in our study.
> >
> > We apologize for not providing a detailed formula to compute this percentage, which may have led to confusion. To address the reviewer's concern, we have added an explanation of this formula in Appendix J of the newly uploaded manuscript. Here is a simplified version for clarity:
> >
> > Given two post-training pruning methods, A and B, with perplexities P(A) and P(B), and the baseline perplexity P(D) of the dense network, the percentage by which method B prevents the performance drop compared to method A can be computed as $\frac{P(A) - P(B)}{P(A) - P(D)}$. This formula provides a fair evaluation of the newly proposed method compared to previous methods, considering the dense model's performance in the post-training pruning field. Using this calculation, the improvement of RIA in preventing performance drops compared to Wanda is 17% across all models, and 50% compared to SparseGPT across all models.
> >
> >
> > **Weakness 3:** At the same time, perplexity increases from the dense baseline are still quite large, especially for 2:4. Hence, it is not clear how useful the corresponding sparse models would be in practice.
> >
> > **Reply:** Thanks for this comment. We admit that in the domain of generative tasks, the application of 2:4 sparsity should be approached with caution since as the reviewer's comments, perplexity increases from the dense baseline are still quite large. However, it is crucial to highlight that in tasks like classification and discrimination, our findings suggest that the N:M model, particularly adopting the channel permutation strategy proposed in this article, can be highly useful and offer a balance between efficiency and performance.
> >
> > **Weakness 4:** Reviewer Comment: There does not appear to be any code in the Supplementary. I hope the authors plan to open-source their work to aid reproducability.
> >
> > **Reply:** Thanks. We will open-source all the codes in this study with the Github link in the camera-ready version article.
> >
> > **Question 1:** Reviewer Comment: How does 4:8 perform with channel permutations in the setup of Table 3?
> >
> > **Reply:** Sorry for the typo. The last column should be 4:8 + CP. We have fixed it in our updated version.
> >
> > **Question 2:** Did you carry out any additional ablation studies around parameter a other than Table 2? I am curious if a = 0.5 is generally the sweet spot or if it was just picked for simplicity.
> >
> > **Reply:** Thanks for the question. To address the reviewer’s concern, we added Figure 10 in the Appendix in the newly uploaded version to explain the reason we chose 0.5 in the manuscript. By comparing the result from different “a” on LLaMA2-7b and LLaMA-13b, a=0.2-0.5 show similar performance. For simplicity, we decided to choose a=0.5. This results are summarized as follow:
> >
> > | a    | 0.1   | 0.2   | 0.3   | 0.4   | 0.5   | 0.6   | 0.7   | 0.8   | 0.9   | 1.0   |
> > |------|-------|-------|-------|-------|-------|-------|-------|-------|-------|-------|
> > | LLaMA2-7b   | 6.923 | 6.820 | 6.798 | 6.802 | 6.806 | 6.799 | 6.815 | 6.847 | 6.862 | 6.882 |
> > | LLaMA-13b   | 6.124 | 6.080 | 6.078 | 6.084 | 6.076 | 6.093 | 6.106 | 6.111 | 6.125 | 6.140 |

---

> > > ### Comment · Reviewer_hn9p · 2023-11-22
> > > **Reviewer Response**
> > >
> > > Thank you for the clarification regarding the 4:8 + CP typo. Unfortunately, this makes the CP results much less impressive. Initially, I thought that 2:4 + CP significantly outperforms 4:8, as indicated by this (incorrectly labeled) column, which I was quite positively surprised by. This now puts the CP improvements in the same 10-20% relative ballpark as the ones of RIA. As acknowledged in my initial review, these are indeed small steps forward, but are unlikely to have a significant impact on the current practicality of LLM pruning by themselves.
> > >
> > > In terms of novelty, I still maintain that RIA and CP are extensions of insights from prior work. This is again reasonable but does not lead to particulary strong conceptual or methodological novelty of the submission.
> > >
> > > Hence, I believe that my initial positive, but not overly so, assessment remains justified.

---

### Official Review · Reviewer_ee73 · 2023-11-01

**Soundness:** 3 good
**Presentation:** 3 good
**Contribution:** 3 good
**Rating:** 6
**Confidence:** 4

**Summary:**

Main Contribution:
- The paper proposes two new methods for efficient post-training pruning of large language models (LLMs):
    1) Relative Importance and Activation (RIA), a new pruning metric that considers both weight and activation information to determine weight importance.
    2) Channel Permutation, a method to maximize retention of important weights when converting a model to N:M sparsity for hardware acceleration.

Novelty:
- RIA provides better pruning performance than prior state-of-the-art methods by avoiding pruning entire channels and using activations to assess weight importance.
- Channel Permutation reformulates the input channel permutation problem as a linear sum assignment problem, allowing efficient optimization using the Hungarian algorithm.

Experiments:
- Experiments conducted on LLMs including LLaMA, LLaMA-2, and OPT ranging from 7B to 70B parameters.
- Tasks: Language modeling (Wikitext-2 perplexity) and zero-shot classification (5 commonsense datasets).
- Compared RIA to magnitude pruning, SparseGPT, and Wanda for unstructured pruning.
- Evaluated Channel Permutation combined with RIA and other methods under N:M sparsity.

Results:
- RIA outperforms prior state-of-the-art post-training pruning methods in both unstructured and N:M sparsity settings.
- Channel Permutation further improves performance under N:M sparsity by efficiently finding better channel arrangements.
- Together, RIA and Channel Permutation provide an effective pipeline for LLM pruning and acceleration with negligible performance loss.

Conclusion:
- RIA and Channel Permutation establish new state-of-the-art results for efficient one-shot post-training pruning of LLMs.
- The proposed methods enable practical acceleration and size reduction of large models.

**Strengths:**

1. Proposes two novel methods (RIA and Channel Permutation) that provide state-of-the-art performance for post-training pruning of large language models.

2. Comprehensive experiments conducted on multiple popular LLMs across a range of model sizes from 7B to 70B parameters.

3. Evaluated on diverse tasks including language modeling and zero-shot classification to demonstrate generalization.

4. Provides both theoretical analysis and empirical results to demonstrate the efficiency and efficacy of the proposed techniques.

5. RIA and Channel Permutation can be readily combined into an effective pipeline for practical LLM pruning and acceleration, with negligible performance loss.

**Weaknesses:**

Overall the manuscript has solid contributions, but expanding the variety of models, tasks, and languages could strengthen the demonstrated effectiveness. Testing scalability and comparing to other recent techniques would also help round out the evaluation. But within the chosen scope, the paper delivers valuable advancements for efficient LLM pruning.

* For "We employ 128 samples from the C4 dataset":
using only 128 samples from C4 as the calibration data is quite limited. With so few samples, the activation statistics may not sufficiently capture the full distribution.

**Questions:**

* what is the channel here in the Transformer models? Transformer models does not have channel or column.

* for ""We employ 128 samples from the C4 dataset", is it possible/worth to do the experiments on a larger and more diverse calibration set (128 might be limited)?

---

> ### Author Response · Authors · 2023-11-21
> **Response to Reviewer 2 (part 1/2)**
>
> Dear Reviewer ee73,
>
> Thank you for acknowledging the contributions of our manuscript and for providing valuable feedback to further enhance its impact. We are pleased to address the highlighted concerns and questions.
>
> **Weakness 1:** Overall the manuscript has solid contributions, but expanding the variety of models, tasks, and languages could strengthen the demonstrated effectiveness. Testing scalability and comparing to other recent techniques would also help round out the evaluation. But within the chosen scope, the paper delivers valuable advancements for efficient LLM pruning.
>
> **Reply:** We thank the suggestion of the reviewer and we agree that we need to test the scalability and comparing to other recent techniques. To address the reviewer's concern, we added experiments on LLM-pruner and pruning only based on the activation which can be seen as a comparison between structured pruning, N:M pruning and unstructured pruning. In addition, we also included the experiments of combining post-training pruning and post-training quantization into consideration. Please refer to the general comments.
>
> **Weakness 2:** For "We employ 128 samples from the C4 dataset": using only 128 samples from C4 as the calibration data is quite limited. With so few samples, the activation statistics may not sufficiently capture the full distribution.
>
> **Reply:** Thank you for your insightful suggestion. In response, we have conducted a comprehensive comparison using varying numbers of samples from 2 to 512. Please refer to Figure 9 in the new uploaded manuscript. The experiments are conducted on LLaMA2-7b. The calibration dataset used in this experiment is C4 and evaluation is assessed on Wikitext2. The sparsity is always fixed at 50%.
>
> PPL changes regarding the increase of number of samples.
>
> |           | 2        | 8        | 32       | 128      | 256      | 512      |
> | --------- | -------- | -------- | -------- | -------- | -------- | -------- |
> | SparseGPT | 7.46     | 6.59     | 6.23     | 6.03     | 5.99     | 5.99     |
> | Wanda     | 6.12     | 6.04     | 6.03     | 5.97     | 5.97     | 5.97     |
> | RIA       | **5.86** | **5.85** | **5.85** | **5.83** | **5.83** | **5.83** |
>
> Pruning time (s) changes regarding the increase of number of samples.
>
> |           | 2    | 8    | 32   | 128  | 256  | 512  |
> | --------- | ---- | ---- | ---- | ---- | ---- | ---- |
> | SparseGPT | 462  | 508  | 790  | 1040 | 2245 | 7842 |
> | Wanda     | 54   | 147  | 156  | 158  | 417  | 1426 |
> | RIA       | 54   | 62   | 138  | 158  | 384  | 1565 |
>
> Our findings indicate that increasing the number of calibration samples does not significantly enhance performance. However, it does lead to a substantial increase in the pruning time of all the methods.
>
> We believe this outcome is due to the distinct nature of calibration samples as compared to training samples. Calibration data is primarily utilized for calculating the activation distribution, rather than for shaping the loss curve. This distinction suggests that a large dataset is not as critical for calibration as it is for training.
>
> Furthermore, this approach aligns with methodologies used in other notable works, such as SparseGPT and Wanda, which also employ 128 calibration data samples.
>
> It's noteworthy that the performance of RIA remains stable as the number of samples increases, achieving a notably low Perplexity (PPL) with as few as two calibration samples.

---

> ### Author Response · Authors · 2023-11-21
> **Response to Reviewer 2 (part 2/2)**
>
> **Questions 1:** what is the channel here in the Transformer models? Transformer models does not have channel or column.
>
> **Reply:** Thank you for your question. In our study, when we refer to 'channels' in the context of Transformer models, we are specifically referring to the dimensions of the input or output weights in the model's layers, which aligns with a concept frequently used in the context of MLP.
>
> In more detail, the term 'channel' in our usage can be understood as analogous to rows or columns of the weight matrices in Transformer layers. For example, in a fully connected layer of an MLP or a linear layer of a Transformer, each row (or column, depending on the implementation) of the weight matrix can be thought of as a 'channel'. These 'channels' represent different learned features or filters that the model applies to the input data.
>
> This terminology, while more commonly associated with CNNs, is also applicable to Transformer models, especially when discussing the dimensions of weights and their transformations. The use of the term 'channel' in this context allows us to discuss the architecture and optimization of these models more precisely, particularly in relation to weight sparsity and efficiency improvements, which are central themes in our research.
>
> Our adoption of this terminology is consistent with established literature, including the NVIDIA article [1], which discusses channel permutations as a method for implementing sparsity in neural networks. It's also important to note that similar terminologies are used in other works in the field [2].
>
> [1] Channel permutations for N: M sparsity, NeurIPS 2021
>
> [2] A simple and effective pruning approach for large language models
>
>
>
> **Question 2:**  for ""We employ 128 samples from the C4 dataset", is it possible/worth to do the experiments on a larger and more diverse calibration set (128 might be limited)?
>
> **Reply:** Thanks for the question. We have done the experiments and included them in the second response.

---

### Official Review · Reviewer_NPvK · 2023-11-05

**Soundness:** 3 good
**Presentation:** 3 good
**Contribution:** 3 good
**Rating:** 6
**Confidence:** 3

**Summary:**

This paper addresses the growing demand for efficient memory and computation in large language models (LLMs). Existing post-training pruning methods have attempted to reduce model size and computation but have not achieved optimal performance. The paper introduces a plug-and-play solution for post-training pruning of LLMs, featuring two innovative components: 1) Relative Importance and Activations (RIA), a novel metric that efficiently considers weight and activations in LLMs, and 2) Channel Permutation, a new approach to maximize the preservation of important weights with N:M sparsity. These components can be combined to enhance N:M structured pruning of LLMs. Empirical experiments demonstrate that RIA alone surpasses existing pruning methods on various LLMs. Moreover, N:M structured pruning with channel permutation can even outperform the original LLaMA2 70B on zero-shot tasks, while providing practical speed-up on specific hardware.

**Strengths:**

- Consider both weights and activations for unstructured pruning in LLMs is novel.
- Particularly, the consideration of relative weight importance is a novel approach.
- Channel permutation is a simple yet effective method for achieving N:M sparsity.

**Weaknesses:**

The reviewer recognizes the novelty and simplicity of the overall approach but has raised substantial concerns. The main issues pertain to the weaknesses in the baselines, which make it challenging for me to be convinced of the effectiveness of the proposed method. Moreover, the motivation and analysis provided appear to be inadequate. For example, concerning the former issue, the following questions come to mind: even if the proposed method can enhance the performance of N:M sparsity-based approaches, are N:M sparsity-based methods genuinely effective? Are they superior to contextual sparsity-based methods?

I describe specific questions and suggestions regarding concerns as follows:

- Insufficient experimental support for motivation: This paper argues the existence of 'channel corruption' asserting that removing input/output channels results in decreased performance as observed in prior works. However, the paper lacks empirical evidence to substantiate this claim. It would be valuable if the authors could include preliminary experiments to provide a basis for their motivation.
- According to AWQ [1], activation-aware weight quantization, which selects important weights based on activation distribution rather than weight distribution, outperforms traditional weight-based quantization. Inspired it, the reviewer suggests that it would be meaningful to consider baseline methods based on activation-based weight pruning for comparison. Therefore, the authors might incorporate and compare unstructured pruning based solely on activations in Table 2.
- In addition to the comparisons with N:M sparsity methods in Table 4 and Table 5, it is advisable to include a comparison with other structured pruning techniques in terms of performance and inference speed improvement. For instance, including a method like Dejavu [2] in the comparison would enhance the comprehensiveness of the evaluation.
- What is the relevance of the experiments in Figure 2 to the claim that activation outliers exist independently of the dataset and model's parts? The reviewer thinks that even if activation values exhibit a high correlation between two datasets, it is possible that activation outliers can be eliminated. Therefore, it would be helpful to clarify the connection between Figure 2 and the claim about activation outliers.
- Can we expect additional performance improvements when combined with post-training quantization methods such as Smooth Quant [3] or AWQ [1]?

[Minor]
- Why is the title "plug-and-play"?
- The hyperlink in the 6-page appendix seems to be incorrect.

[1] AWQ: Activation-aware Weight Quantization for LLM Compression and Acceleration

[2] Deja Vu: Contextual Sparsity for Efficient LLMs at Inference Time, ICML 2023

[3] SmoothQuant: Accurate and Efficient Post-Training Quantization for Large Language Models, ICML 2023

**Questions:**

Please address the concerns in Weaknesses

---

> ### Author Response · Authors · 2023-11-21
> **Response to Reviewer 1 (part 1/3)**
>
> Dear Reviewer NPvK,
>
> Thank you for your review of our paper. We appreciate your positive evaluation and also the questions to our submission which aims at improving our presentation and encouraging us to process more empirical evidence to support our methods.
>
> We will reply to each of the weakness you mentioned and the questions you raised.
>
> **Weakness 1:** Insufficient experimental support for motivation: This paper argues the existence of 'channel corruption' asserting that removing input/output channels results in decreased performance as observed in prior works. However, the paper lacks empirical evidence to substantiate this claim. It would be valuable if the authors could include preliminary experiments to provide a basis for their motivation.
>
> **Reply:** Thank you for your feedback. We understand the need for empirical evidence to support our assertion of 'channel corruption' and have taken steps to address this in our revised manuscript Appendix E (Evidence of Channel Corruption).
>
> **1. Evidence of Channel Corruption:** We have updated our manuscript to include new empirical data demonstrating channel corruption. As illustrated in Figure 6, our analysis of the Wanda [1] model shows that approximately **10% of channels exhibit corruption**. This evidence directly supports our claim and provides a clearer understanding of the phenomenon.
>
> **2. Detrimental Impact of channel corruption on Model Performance:**
>
> - We have conducted experiments using LLM-Pruner [2] which is a SOTA structured pruning method used in LLMs. We also compare our method with the baseline structured pruning method of pruning merely based on the activation aligning with the experiment you proposed in **Weakness 2**. We utilize this comparison to further illustrate the effects of channel corruption.
>
> - We separated the experiments to two different Tables. One utilized the strategy in LLM-Pruner that only pruned the layers from 4-30 layers and the other one pruned all the layers.
>
> - Our results indicate: 1) The performance of LLM-Pruner, perplexity, notably increases at already a low sparsity level of 25%. 2) Using LLM-Pruner's approach, which involves skipping 6 out of 32 layers, led to lower perplexity scores (PPLs). However, the balance between performance and the increased inference speed due to sparsity requires further exploration. We aim to continue investigating this issue promptly for this study.
>
> **3. Additional Section in Appendix E:** To further address your concerns, we have added a dedicated section in Appendix E of our manuscript. This section explicitly discusses the influence of channel corruption on post-training pruning methods. Due to time constraints, our current experiments are limited to the LLaMA2-7b model. We plan to extend our investigation to more models to provide comprehensive evidence of channel corruption and its implications.
>
> **Prune from layer 4 to layer 32**
>
> |                               | 1:4 (25% sparsity) | 2:4 (50% sparsity) | 3:4 (75% sparsity) |
> | ----------------------------- | ------------------ | ------------------ | ------------------ |
> | LLM-Pruner (structured)       | 26.04              | 77.90              | 464.34             |
> | Activation-based (structured) | 30.23              | 525.5              | 8692.2             |
> | Wanda (semi-structured)       | 5.84               | 9.21               | **432.6**          |
> | RIA (semi-structured)         | 5.82               | 8.91               | 442.6              |
> | RIA + CP (semi-structured)    | **5.74**           | **8.03**           | 436.1              |
>
> **Prune all the layers**
>
> |                               | 1:4 (25% sparsity) | 2:4 (50% sparsity) | 3:4 (75% sparsity) |
> | ----------------------------- | ------------------ | ------------------ | ------------------ |
> | LLM-Pruner (structured)       | 28.88              | nan                | 13570              |
> | Activation-based (structured) | 39.54              | 10467              | nan                |
> | Wanda (semi-structured)       | 5.94               | 12.15              | 2863.3             |
> | RIA (semi-structured)         | 5.90               | 11.27              | 1891.13            |
> | RIA + CP (semi-structured)    | **5.81**           | **10.12**          | **1532.72**        |

---

> ### Author Response · Authors · 2023-11-21
> **Response to Reviewer 1 (part 2/3)**
>
> **Weakness 2:** According to AWQ [1], activation-aware weight quantization, which selects important weights based on activation distribution rather than weight distribution, outperforms traditional weight-based quantization. Inspired it, the reviewer suggests that it would be meaningful to consider baseline methods based on activation-based weight pruning for comparison. Therefore, the authors might incorporate and compare unstructured pruning based solely on activations in Table 2.
>
>
>
> **Reply:**
> - Thanks for the question. Pruning solely on activation means the whole channel will be removed together, thus pruning on activation actually belongs to structured pruning. We have applied this method on **LLaMA2-7b, LLaMA1-13b, and LLaMA1-30b**. The results are included in the Table on the last response and also in updated **Table 2** as an ablation test in the newly uploaded version of the manuscript.
>
> - The results indicate that relying solely on activation, the PPL **increases significantly** with just 25% sparsity, reaching **over 5000 at 50% sparsity** on each model. This suggests that **structured pruning is not effective** in post-training pruning scenarios, as the pruned information from the entire channels cannot be recovered without retraining or fine-tuning.
>
>
>
> **Weakness 3:** In addition to the comparisons with N:M sparsity methods in Table 4 and Table 5, it is advisable to include a comparison with other structured pruning techniques in terms of performance and inference speed improvement. For instance, including a method like Dejavu [2] in the comparison would enhance the comprehensiveness of the evaluation.
>
>
>
>
> **Reply:**
>
> - To enhance the comprehensiveness of our evaluation, we have included additional results from the **LLM-pruner [2]**, a **state-of-the-art structured pruning** method in our article, as detailed in our first response. These results demonstrate the superiority of our N:M semi-structured pruning.
>
>
> - As for **Dejavu [3]**, we would like to clarify that **Dejavu** cannot be fairly considered for comparison in this study because it differs fundamentally from our method in terms of its **operational framework**.
>
>   - **Dejavu** is not a post-training pruning method as it introduces **additional parameters** and **requires the predictor to be trained for several epochs** (50, as per the provided code).
>   - Consequently, **Dejavu** requires computational time resources, while our method is designed to be applied **without retraining, offering a 'plug-and-play' solution** for practical scenarios.
>
> - To address this distinction and your concern, we have added the following statement in the related work section of our manuscript.
>
>   “Note that the method proposed in this article is designed for application without retraining and finetuning, which differentiates it from techniques like Dejavu [3] that require additional training steps. Consequently, our comparisons are focused on methods that do not involve retraining, such as Magnitude, Wanda, and SparseGPT.”
>
> - Regarding inference speed improvement, we offered the **acceleration rate** of the semi-structured pruning in Section 5.4. For the end-to-end inference acceleration of structured sparsity and semi-structured sparsity, we here offer their inference latency. Our experiment is on LLaMA2-7b, the sequence length of the input is 12. The deployment is on 2 Nvidia A100 80GB in parallel on Ubuntu 22.04.1 system. Here, we offer the inference latency with varying batch sizes.
>
>   |                         | 1         | 8         | 16        | 64         |
>   | ----------------------- | --------- | --------- | --------- | ---------- |
>   | dense                   | 281.13 ms | 547.82 ms | 1011.81ms | 3742.76 ms |
>   | structured 50% sparsity | 238.92 ms | 326.14 ms | 616.13 ms | 2181.69 ms |
>   | 2:4 sparsity            | 225.60 ms | 357.48 ms | 731.81 ms | 2495.41 ms |
>
>   The table shows that structural 50% sparsity marginally outperforms the 2:4 sparsity model as batch sizes increase. It's important to note that when dealing with smaller input batches, both structured 50% sparsity and 2:4 sparsity models do not significantly boost inference speed. However, as the batch size grows, the acceleration for 2:4 sparsity approximates 1.5x, whereas the structured 50% sparsity tends to reach about 1.7x acceleration.

---

> ### Author Response · Authors · 2023-11-21
> **Response to Reviewer 1 (part 3/3)**
>
> **Weakness 4:** What is the relevance of the experiments in Figure 2 to the claim that activation outliers exist independently of the dataset and model's parts? The reviewer thinks that even if activation values exhibit a high correlation between two datasets, it is possible that activation outliers can be eliminated. Therefore, it would be helpful to clarify the connection between Figure 2 and the claim about activation outliers.
>
> **Reply:**
> - Thanks for the reviewer raising this question. To address the reviewer’s concern, we added the below content in Section 3.3 in the updated file.
>
>   “We offer evidence that the Spearman Rank correlation between pairs of the activations of different datasets is positive. This positivity is a necessary condition to incorporate the activation into our RIA formula. And indeed it is always satisfying.”
>
> **Weakness 5:** Can we expect additional performance improvements when combined with post-training quantization methods such as Smooth Quant or AWQ?
>
> **Reply:**
> - Thank you for your insightful question. Because there is not a remarkable decrease in performance using quantization, it indicates that our method could also be applied in combination with quantization. This will be a new and valuable point to comment on in the article.
>
>
> - We tested on two quantization methods: **GPTQ [4]** and **AWQ [5]**. Here are the results for pruning + GPTQ and pruning + AWQ on LLaMA2-7b, weight bit (wbit) = 4, using C4 for calibration and Wikitext2 for evaluation. We still get the **best performance** on two quantizations among all pruning methods:
>
>   |                                       | Magnitude | SparseGPT | Wanda | RIA      |
>   | ------------------------------------- | --------- | --------- | ----- | -------- |
>   | Unstructured (50% sparsity)           | 16.02     | 6.99      | 6.92  | **6.81** |
>   | Unstructured (50% sparsity) With GPTQ | 15.21     | 7.59      | 7.41  | **7.28** |
>   | Unstructured (50% sparsity) With AWQ  | 17.12     | 7.15      | 7.10  | **6.97** |
>
>
>
> - We have identified two primary strategies for merging post-training pruning with quantization: first pruning then quantizing (a), and first quantizing then pruning (b.)
>   - Our findings indicate **a preference for (a) pruning before quantizing.**
>   - This is potentially because for (b), conventional block-wise quantization relies merely on the min and max weights within a block, and pruning can lead to a reduction in these extremes, thereby potentially aiding quantization.
>   -  In contrast, (a) quantizing prior to pruning adversely affects the computation of weight importance for pruning, resulting in diminished performance.
>
> - The discussion is added as Appendix I in our manuscript.
>
>
>
>
> **Minor:** **Why is the title "plug-and-play"?**
>
> **Reply:**
>
> - The reason we call it plug-and-play is because of 3 reasons.
>
>   - Ignorable Pruning time.
>   - No need for any additional finetuning or retraining.
>   - With channel permutation, our method can perform well in the zero-shot experiments when adopting N:M sparsity pattern while currently other post-training pruning methods cannot.
>
> - To further address the reviewer’s concern, we introduce in the discussion: “We decide to call our method “plug-and-play” because, in the zero-shot experiment, it demonstrated performance similar to the original dense model with no need for any additional finetuning or retraining whereas, the other post-training pruning method such as Sparsegpt and Wanda are not “Plug-and-play” because they don’t have channel permutation.”
>
> **Minor:** **The hyperlink in the 6-page appendix seems to be incorrect.**
>
> **Reply:** Thank you. We have fixed that in our updated version of the paper.
>
>
>
> Reference:
>
> [1] Optimal Brain Compression: A Framework for Accurate Post-Training Quantization and Pruning, NeurIPS 2022
>
> [2] SparseGPT: Massive Language Models Can be Accurately Pruned in One-Shot, ICML 2023
>
> [3] LLM-Pruner: On the Structural Pruning of Large Language Models, NeurIPS 2023
>
> [4] GPTQ: ACCURATE POST-TRAINING QUANTIZATION FOR GENERATIVE PRE-TRAINED TRANSFORMERS, ICLR 2023
>
> [5] AWQ: Activation-aware Weight Quantization for LLM Compression and Acceleration

---

> > ### Comment · Reviewer_NPvK · 2023-11-22
> > **Response by reviewer**
> >
> > Thank you for the response and extensive experiments. Most critical concerns are addressed and I would like to keep my rating as weak accept.
> >
> > I have additional minir questions regarding W3 and W4.
> >
> > Regarding W3, while Dejavu trains the predictor, it do not need to retrain LLMs directly. Also, the predictor is just MLP layers, so 50 epochs predictor training time might be just a few GPU minutes and required computing resource is negligible. Thus, the reviewer still think that we can regard Dejavu as the post-training pruning since we do not retrain LLMs.
> > Can the authors conduct experiments to compare Dejavu and the proposed method and discuss the strength of N:M sparsity-based methods at the inference stage in terms of inference time and performance compared with context-based pruning like Dejavu?
> > Considering Dejavu use additional components and resources, the experimental results will not affect to rate the proposed method.
> >
> > Regarding W4, It is still difficult for me to understand. Why a positive correlation between two datasets is the necessary condition to include activation values to RIA formula? Could the authors explain more about it? Also, then does outliers no more related with RIA formula?
> >
> > Thanks, reviewer.

---

> > > ### Author Response · Authors · 2023-11-22
> > > **Response to the further questions**
> > >
> > > Dear Reviewer,
> > >
> > > Thanks for your fast reply. We are thankful for these two additional questions.
> > >
> > > Here are the replies:
> > >
> > > **Q1.** Regarding W3, while Dejavu trains the predictor, it do not need to retrain LLMs directly. Also, the predictor is just MLP layers, so 50 epochs predictor training time might be just a few GPU minutes and required computing resource is negligible. Thus, the reviewer still think that we can regard Dejavu as the post-training pruning since we do not retrain LLMs. Can the authors conduct experiments to compare Dejavu and the proposed method and discuss the strength of N:M sparsity-based methods at the inference stage in terms of inference time and performance compared with context-based pruning like Dejavu? Considering Dejavu use additional components and resources, the experimental results will not affect to rate the proposed method.
> > >
> > > **Reply:** Thank you for the reminder. It's true that there isn't a clear, universally accepted definition of post-training pruning. As pointed out by the reviewer, the additional training time and computational resources required for Dejavu are minimal. Therefore, we concur with the reviewer's view that Dejavu qualifies as a post-training pruning method. We have updated our earlier discussion of Dejavu in the related work section to reflect this perspective. <<Some methods, such as Dejavu, introduce extra parameters and require an additional training step, but the time and resources needed for this are ignorable. Hence, we consider them also as a post-training pruning method. We have included Dejavu as the representative in our comparison in Appendix E, which focuses on contrasting N:M constraint and structured pruning.>>
> > >
> > > We are now conducting comprehensive experiments to compare semi-structured RIA with Dejavu. Hope we can catch up with the deadline.
> > >
> > >
> > > **Q2.** Regarding W4, It is still difficult for me to understand. Why a positive correlation between two datasets is the necessary condition to include activation values to RIA formula? Could the authors explain more about it? Also, then does outliers no more related with RIA formula?
> > >
> > > **Reply:** Thank you for your question, which prompts us to clarify our motivation more clearly.
> > >
> > > Our paper explores two primary reasons for integrating activation into our formula:
> > >
> > > - The Presence of Outliers: As discussed in the paper, outliers play a significant role in our consideration.
> > > - The Impact of Pruning Across Input Channels: Pruning is not limited to individual input channels, so it's insufficient to only focus on outliers. It's crucial to establish a positive correlation between activations across different datasets. For instance, a negative correlation between activations suggests inconsistent activation distributions across datasets. To illustrate, consider two activations, a1 and a2. In one dataset, a1 might consistently exceed a2, while in another, a2 surpasses a1. If we incorporate these activations into the pruning process, this could result in preserving more weights of a1 channel when using the first dataset as calibration, however, a2 is more important in dataset 2. Therefore, assessing the correlation is a vital preliminary step before integration.
> > >
> > > We thank the reviewer for raising this question and we revised the sentences in the article to make it clear.
> > > <<We offer evidence that the Spearman Rank correlation between pairs of the activations of different datasets is positive. A positive correlation indicates a relatively consistent activation pattern across different datasets, reinforcing the reliability of incorporating the activation information of our approach in diverse data environments.>>
> > >
> > > Please let us know whether we explain it clearly.
> > >
> > > Thanks,
> > >
> > > Authors

---

### Author Response · Authors · 2023-11-21
**General Comments and Updated Manuscript**

We express our gratitude to all reviewers for their detailed evaluations, valuable insights, and constructive criticisms. Their contributions have significantly enhanced the depth and clarity of our analysis in the article.

In response to the reviewers' queries, we have conducted comprehensive experiments and made extensive revisions to the manuscript. All key revisions, as discussed with the reviewers, are highlighted in red for ease of identification. Below, we summarize the principal modifications made in light of the reviewers' recommendations:

1. Appendix E now includes a detailed explanation of the **channel corruption phenomenon** observed in the post-training pruning scenario.

2. In Appendix E, we present new experiments comparing the performance of N:M sparsity with **structured pruning**, specifically focusing on the **LLM-pruner**.

3. We have added results of **end-to-end inference latency**, comparing N:M sparsity with structured sparsity, in Appendix E.

4. Appendix E also included experiments exploring **various levels of sparsity** under the N:M constraint.

5. To address the integration of post-training pruning methods with **quantization** techniques, we introduced Appendix I, which covers **GPTQ** and **AWQ** methods.

6. We added Appendix A to provide **empirical observations** which **motivates** us to propose the channel permutation method.

7. To enhance the reliability of our comparisons with Wanda and Sparsegpt, we have **increased the number of calibration samples** from 128 to 512 in Figure 9.

8. New experiments have been conducted to determine **the optimal selection of the factor 'a'** in the RIA formula, with findings presented in Figure 10.

These changes, we believe, will comprehensively address the concerns and suggestions raised by the reviewers, thereby enriching the overall quality and impact of our research.

---

### Meta-Review · Area_Chair_1aWv · 2023-12-09

**Metareview:**

The paper proposes a method for pruning a trained LLM using two ingredients: RIA that address the issue of full-channel pruning by using a relative importance measure, and channel permutation that results in improved performance with structured N:M sparsity. Authors have added several experiments during the rebuttal phase which have addressed the comments from reviewers. All reviewers agree that the paper proposes an advance in the relevant problem of post-training pruning.

**Justification For Why Not Higher Score:**

While the paper shows that RIA and channel permutation leads to improved pruning performance, the ideas are an extension of earlier works in the literature and the empirical gains are relatively small (pointed out by reviewer hn9p).

**Justification For Why Not Lower Score:**

All reviewers acknowledge the paper has sufficient contributions for acceptance.

---

### Decision · Program_Chairs · 2024-01-16

Accept (poster)